



# FYRE Climate: A high-resolution reanalysis of daily precipitation and temperature in France from 1871 to 2012

Alexandre Devers[1], Jean-Philippe Vidal[1], Claire Lauvernet[1], and Olivier Vannier[2]

[1]INRAE, UR RiverLy, 5 rue de la Doua, CS 20244, 69625 Villeurbanne Cedex, France
[2]Compagnie Nationale du Rhône (CNR), 2 rue André Bonin, 69004 Lyon, France

**Correspondence:** Jean-Philippe Vidal (jean-philippe.vidal@inrae.fr)

**Abstract.**

Surface observations are usually too few and far between to properly assess multidecadal variations at the local scale and characterize historical local extreme events at the same time. A data assimilation scheme has been recently presented to assimilate daily observations of temperature and precipitation into downscaled reconstructions from a global extended reanalysis through an Ensemble Kalman fitting approach and derive high-resolution fields. Recent studies also showed that assimilating observations at high temporal resolution does not guarantee correct multidecadal variations. The current paper thus proposes (1) to apply this scheme over France and over the 1871-2012 period based on the SCOPE Climate reconstructions background dataset and all available daily historical surface observations of temperature and precipitation, (2) to develop an assimilation scheme at the yearly time scale and to apply it over the same period and lastly, (3) to derive the FYRE Climate reanalysis, a 25-member ensemble hybrid dataset resulting from the daily and yearly assimilation schemes, spanning the whole 1871-2012 period at a daily and 8-km resolution over France. Assimilating daily observations only allows reconstructing accurately daily characteristics, but fails in reproducing robust multidecadal variations when compared to independent datasets. Combining the daily and yearly assimilation schemes, FYRE Climate clearly performs better than the SCOPE Climate background in terms of bias, error, and correlation, but also better than the Safran reference surface reanalysis over France available from 1958 onward only. FYRE Climate also succeeds in reconstructing both local extreme events and multidecadal variability. It is made freely available from http://doi.org/10.5281/zenodo.4005573 (precipitation) and http://doi.org/10.5281/zenodo.4006472 (temperature).

## 1 Introduction

Several studies show that long-term meteorological observation often display strong multidecadal variations both in terms of annual values (Slonosky, 2002) and extremes (Willems, 2013). These variations in meteorological variables end up affecting multidecadal variations of discharge observations (Boé and Habets, 2014). However, the few available long-term observations do not allow to grasp the evolving climate in a spatially continuous way. To solve this discontinuity issue, daily meteorological high-resolution surface reanalyses have been built at the country scale (Vidal et al., 2010a; Quintana-Segui et al., 2017) or spanning Europe (Landelius et al., 2016; Soci et al., 2016). These reanalyses are mainly built using Optimal Interpolation



(Gandin, 1965) combining daily observations and large-scale atmospheric reanalyses as background. However, due to the low number of daily meteorological observations before the 1950s (Caillouet et al., 2019), these reanalyses are usually limited to the second half of the Twentieth Century (Minvielle et al., 2015). This lack of sufficient daily historical observations in many countries in Europe led to the creation of several long-term high-resolution reconstructions. These datasets are mainly built using statistical downscaling of global atmospheric reanalyses (Dayon et al., 2015; Minvielle et al., 2015; Caillouet et al., 2019;

Horton and Brönnimann, 2018), but in data rich areas, some are also built as an interpolation of surface observations (Keller et al., 2015).

In the past few years, some studies have also developed or used different processes to take advantage of both historical observations and downscaled reconstructions. For instance, the downscaled reconstructions may be modified using individual long-term observed times series (Kuentz et al., 2015; Brigode et al., 2016). Observations may also be integrated in a post-

processing of the downscaling step, through e.g. the selection of a unique member among a downscaled ensemble (Bonnet et al., 2017, 2020; Minvielle et al., 2015).

In parallel, paleoclimate studies that usually deal with coarser temporal and spatial resolutions have used data assimilation (DA) to reconstruct past climate fields. DA usually combines (i) a background, (ii) observations, (iii) a model, and (iv) their associated uncertainty to provide an optimal analysis and its associated error (Asch et al., 2016). DA is usually composed of

two steps: the analysis, and the forecast, which is a propagation of the analysis by the (dynamical) model. In paleoclimate studies in which the propagation step may be highly computationnaly demanding typically, DA methods have been applied "offline" (Goosse et al., 2006; Annan and Hargreaves, 2012; Bhend et al., 2012; Hakim et al., 2016; Valler et al., 2019): the background is computed by the dynamical model once for the entire period, and the DA comes down to the analysis step (Matsikaris et al., 2015).

Some recent studies have also attempted to follow the offline DA methodology at higher resolution to assimilate daily observation into various reconstructions. Devers et al. (2020a) developed a DA scheme of daily precipitation and temperature over France into the SCOPE Climate downscaled reconstruction dataset (Caillouet et al., 2019). They evaluated the performance of the DA scheme over a recent period (2009-2012) by testing several observation densities, and they showed a positive effect even when only few observations representative of the 1870s network density are assimilated. The DA method is an offline

Ensemble Kalman Filter (Evensen, 2003), also referred to as Ensemble Kalman fitting (EnKf, Bhend et al., 2012). Pfister et al. (2020) have also assimilated daily temperature over Switzerland into a statistical reconstruction, leading to an improvement using only a limited number of stations – 25 over the entire Switzerland. However, assimilating observations at a high temporal resolution as in the two previous examples does not guarantee a correct multidecadal variation in the reanalysis (Steiger and Hakim, 2016). To bypass this problem, some studies in paleoclimatology assimilate temporal averages of observations through

offline DA in existing reconstructions (Steiger and Hakim, 2016; Dirren and Hakim, 2005; Huntley and Hakim, 2010; Steiger et al., 2014).

This paper builds upon the work of Devers et al. (2020a) and applies the scheme they developed over the 1871-2012 period to produce the FYRE Daily reanalysis, composed of 25 members of daily precipitation and temperature at a 8 km resolution over France. A new DA scheme at the yearly time scale is then proposed using once again SCOPE Climate as a background.





The scheme is then applied over the 1871-2012 period for both precipitation and temperature leading to the 25-member yearly reanalysis of precipitation and temperature at 8 km resolution: FYRE Yearly. In order to include both multidecadal variations and extreme events, FYRE Daily and FYRE Yearly are hybridized to build a new reanalysis: FYRE Climate. Finally, the benefits of hybridization is assessed by comparing FYRE Daily and FYRE Climate against several products over a recent period (1950-2000) and over the entire 20th century. These comparisons include the computation of several metrics: Continuous Ranked Probability Score (CRPS, Brown, 1974), bias, error, and correlation. The multidecadal variations of the two reanalyses and the background are also compared with those of other products. Furthermore, the reconstruction of extreme events is investigated with the study of an extreme rainfall event during September 1890 and the unusually cold month of December 1879.

The paper is organised as follows: Section 2 introduces the background, the assimilated observations and their metadata, as well as validation datasets. Section 3 describes the DA implementation and the creation of the different reanalyses. Their validation through different comparisons and examples is presented in Sect. 4. Finally, several points are discussed in Sect. 5 and conclusions are drawn in Sect. 6.

## 2 Data

### 2.1 Background

The SCOPE (Spatially COherent Probabilistic Extension Method Caillouet et al., 2016, 2017) method is based on the analogue approach, which assumes that similar atmospheric situations lead to similar local situations (Lorenz, 1969). This method uses an ensemble statistical downscaling to reconstruct local climate fields from large-scale atmospheric situations. To that end it uses 6 analogy levels – see Caillouet et al. (2016, 2017) for their description – through a stepswise approach (Radanovics et al., 2013; Ben Daoud et al., 2016; Caillouet et al., 2016). Finally, two other steps were included by Caillouet et al. (2017): (1) a bias correction based on the Safran reanalysis – see Sect. 2.4.1 below for details about Safran –, and (2) an adaptation of the Schaake Shuffle (Clark et al., 2004) to obtain spatial and inter-variable coherence for each ensemble member.

The application of the SCOPE method using the ensemble mean values of the Twentieth Century Reanalysis (Compo et al., 2011) as a source of large-scale information – predictors – and the Safran reanalysis as an archive for analogues – predictands – has led to the creation of the SCOPE Climate dataset (Caillouet et al., 2019). This daily 25-member ensemble reconstruction is available on a 8 km grid (see Figure 1) over the 1871-2012 period for precipitation (Caillouet et al., 2018a), temperature (Caillouet et al., 2018b), and Penman-Monteith reference evapotranspiration (Caillouet et al., 2018c). Note that as SCOPE Climate resamples Safran data, the daily temperature is actually computed as the daily average of hourly temperature.

The comparison of SCOPE Climate with the independent Météo-France long-term homogenized series (Moisselin et al., 2002, see Sect. 2.4.2 for details about this dataset) have put forward a low and steady error – at the monthly time scale – over the whole 20th century (Caillouet et al., 2019).

Two background ensembles were extracted from SCOPE Climate for this study :



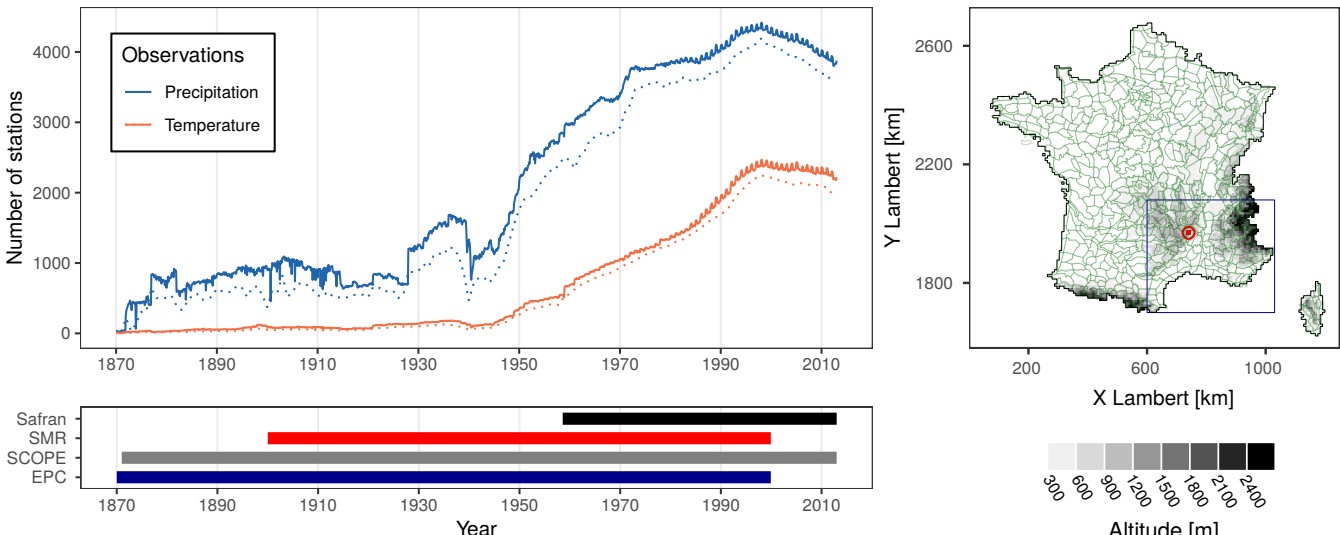

**Figure 1.** Top-left panel: Availability of meteorological observations in the Météo-France database over the 1871–2012 period. Full lines represent the number of open stations during the considered year and dotted lines represent the number of stations with a complete series. Lower-left panel: availability of the different gridded datasets: SCOPE Climate, Safran, European Pattern Climatology, and Monthly homogenized series. Right-panel: map of the 608 climatologically homogeneous zones – in green – as defined in Safran, and altitude of the 8 km grid cells of SCOPE Climate and Safran. The case study cell (id: 7548), highlighted in red, is located in the Cevennes area at the altitude of 1,165m a.s.l. in a mountainous Mediterranean climate. An observation station (id: 7154005, Mazan-l'Abbaye) is located in this cell at the altitude of 1240m a.s.l. The blue frame represents the study area for the heavy rainfall event of 21 September 1890 (see Sect. 4.5)

- for daily DA, the 25-member ensemble of daily values of temperature and precipitation from SCOPE Climate;

- for yearly DA, the 25-member ensemble of yearly-average temperature and the yearly-accumulated precipitation from SCOPE Climate.

In both cases, data were extracted between 1 January 1871 and 29 December 2012, i.e the entire period of availability of SCOPE Climate. 2012 yearly values are computed over the available period.

## 2.2 Assimilated observations

Surface observations originate from the Météo-France database composed of the daily sum of precipitation, and daily minimum and maximum temperature. The observation network has evolved from less than 10 stations of temperature and precipitation
in the 1870s to more than 2500 stations for temperature and 4300 for precipitation at the end of the 20th century (Figure 1, top left). The number of stations with a full year of available data evolved in parallel. This large number of observations is partially based on a strong voluntary observation network in France (Galliot, 2003; Capel, 2009).

Variables used as observations over the 1871-2012 period are :



- for daily DA: the daily sum of precipitation and the daily mean temperature;

– for yearly DA: the yearly sum of precipitation and yearly mean temperature for stations with a full year of available data. Values are otherwise discarded and not used in the yearly DA.

Note that because of the low availability of hourly measurements in the past, the daily mean temperature is here computed as the mean of the daily maximum temperature and the daily minimum temperature. Observed yearly values for 2012 are computed between 1 January to 29 December to stick to the background data availability.

## 110   2.3    Metadata for observations

Along with observed values, some metadata are available over the 1871-2012 period. Three types of metadata have been used in this study in order to define at best the measurement error of temperature and precipitation.

The first type of metadata available over the entire period is the type of station, ranging from 0 for the highest quality to 5 for the lowest quality. This classification is not linked to any numerical values of measurement error but can be used as an 115   indicator of the overall quality of the station. The second type of metadata, noted $\sigma_{MP}$, is only available from 1999 onward and represents the maintained performance of each station (Leroy, 2010, Table 1a). This classification includes the intrinsic quality of the measurement device and the quality of the measurement method. Lastly, the site representativeness, noted $\sigma_{SR}$, is also available over the 1999-2012 period (Leroy and Lèches, 2014, Table 1b). This classification takes into account the error due to the influence of the nearby environment of the station. The maintained performance and site representativeness give 120   information about the daily error measurement and are related to the station quality as established by Météo-France and the World Meteorological Organization (WMO, 2014).

## 2.4    Other datasets

### 2.4.1    Safran

The SAFRAN system is an analysis system based on an Optimal Interpolation scheme that merges in-situ observation (temper-125   ature, precipitation, relative humidity, wind speed, and cloudiness) and a background – ERA-40 large-scale reanalysis (Uppala et al., 2005) and ECMWF operational analyses, or climatological values (for precipitation). The analysis is performed on 608 climatologically homogeneous zones (see Figure 1, top right) and is afterwards disaggregated onto 8602 cells in France (8 km grid) based on altitude only (Quintana-Segui et al., 2008). The Safran reanalysis is available from 1 August 1958 onwards and is updated annually (Vidal et al., 2010b). In this study, daily precipitation and daily temperature – computed as the average 130   of hourly values – are extracted from the Safran database over the 1 January 1958 - 29 December 2012 period. The Safran reanalysis is used here to asses features of the background and the different reanalyses over the last 50 years or so.



**Table 1.** Station classifications and related measurement uncertainty – standard deviation, computed as half of the 95 % confidence interval – for daily temperature and daily precipitation.

(a) Maintained performance (Leroy, 2010). The standard deviation for precipitation is selected as the maximum between the percentage value and the minimum value indicated in brackets (in mm).

|  | A | B | C | D | E |
|---|---|---|---|---|---|
| Temperature [°C] | 0.1 | 0.25 | 0.5 | 0.75 | 1 |
| Precipitation [%] | 2.5 | 2.5 | 5 | 7.5 | 10 |
|  | (0.05) | (0.1) | (0.25) | (0.375) | (0.5) |

(b) Representativeness of the site (Leroy and Lèches, 2014)

|  | 1 | 2 | 3 | 4 | 5 |
|---|---|---|---|---|---|
| Temperature [°C] | 0 | 0 | 0.5 | 1 | 2.5 |
| Precipitation [%] | 0 | 2.5 | 7.5 | 12.5 | 50 |

### 2.4.2 Monthly homogenized series (SMR)

The monthly homogenized series – called SMR for "Séries Mensuelles de Référence" – are produced by Météo-France. The homogenization is intended to detect and correct potential homogeneity breaks related to changes in location or instrumentation (Moisselin et al., 2002; Gibelin et al., 2014). SMR comprise two differents datasets:

- 1583 times series for precipitation and 308 for temperature allowing to capture the spatial patterns over France but only covering the period 1959-2009 (Gibelin et al., 2014);

- 332 times series for precipitation and 88 for temperature covering the period 1900-2000 (Moisselin et al., 2002). Although stations are not distributed in a homogeneous way over France, these series constitute a reliable reference for analyzing multidecadal variations as well as long-term trends.

The SMR is a high-quality dataset that will be used to asses the quality of the background and several reanalyses in terms of multidecadal variations and trends, but also to evaluate their quality at the monthly time scale through different metrics – bias, correlation, and error – over different periods.

### 2.4.3 European Pattern Climatology

The monthly gridded reconstructions of precipitation and temperature developed by Casty et al. (2005, 2007) – here called European Pattern Climatology (EPC) – were created by regressing a network of station data against a modern climate dataset





(CRU TS2, Mitchell and Jones, 2005). Transfer functions via principal component regressions are computed over a recent period where both products are available. Finally, the transfer functions are fed by a limited number of precipitation and temperature stations having a long instrumental record. The reconstruction covers the 1766-2000 period over the North Atlantic/European sector with a spatial resolution of 0.5°. It is important to note that this methodology assumes a stationary behaviour during the entire period. Furthermore, non-homogeneity may appear because the dataset is composed of the CRU TS2 between 1901 and 2000 and of a climate field reconstruction based on principal component regression before 1900.

For this study, values are extracted over the 254 cells covering the France area between January 1871 and December 2000 for precipitation and temperature. The EPC reconstruction will be used to evaluate the coherence of the multidecadal variations of the background and the reanalyses over a long period.

## 3    Data assimilation setup

### 3.1    Ensemble Kalman fitting

The Ensemble Kalman Filter is a sequential data assimilation method relying on an approximation of the Kalman filter in which the error statistics are computed from an ensemble of members (Evensen, 2003). The background is generally computed from a propagation (by the dynamical model) of the analysis state ensemble at the previous time step. In an offline approach such as in this study, only the analysis step is carried out. Hence, we name this application Ensemble Kalman fitting (EnKf) in lieu of Ensemble Kalman Filter (Bhend et al., 2012; Franke et al., 2017).

The background ensemble is noted $\boldsymbol{X}^b \in \mathbb{R}^{n \times N}$ with $n$ the size of the background state vector – i.e the number of grid points – and $N$ the number of ensemble members. In a Gaussian context the background can be defined by the ensemble mean $\overline{\boldsymbol{x}}^b \in \mathbb{R}^n$, and the background error covariance matrix $\boldsymbol{P}^b \in \mathbb{R}^{n \times n}$. In the EnKf, $\boldsymbol{P}^b$ is estimated using the ensemble perturbation matrix $\boldsymbol{X}'_b \in \mathbb{R}^n$:

$$\hat{\boldsymbol{P}}^b = \frac{\boldsymbol{X}'_b \boldsymbol{X}'^T_b}{N-1} \quad \text{with} \quad \boldsymbol{X}'_b = \boldsymbol{X}^b - \overline{\boldsymbol{x}}^b \tag{1}$$

The observation vector $\boldsymbol{y} \in \mathbb{R}^m$ contains all observations (in this case $m$) for a specific time step, that is, in this study, daily or yearly, with an error assumed to be gaussian. Burgers et al. (1998) showed the benefits of perturbed observations in EnKF and demonstrated that using non-perturbed observations can lead to filter divergence (Houtekamer and Mitchell, 1998). Following them, the perturbed ensemble observation matrix $\boldsymbol{Y} \in \mathbb{R}^{m \times N}$ is generated :

$$\boldsymbol{Y} = \boldsymbol{y} + \boldsymbol{\epsilon} \tag{2}$$

where the matrix $\boldsymbol{\epsilon}$ is the ensemble of perturbations $\boldsymbol{\epsilon_i} \in \mathbb{R}^{m \times N}$, for $i = 1, ..., N$ drawn from a normal distribution $\mathcal{N}(0, \boldsymbol{\sigma}^2_{\mathrm{obs}})$, with $\boldsymbol{\sigma}^2_{\mathrm{obs}}$ the observation error variance.





The analysis step of the Ensemble Kalman Filter can be solved using the two following equations from the original Kalman Filter:

$$
\begin{cases}
\boldsymbol{X}^a = \boldsymbol{X}^b + \hat{\boldsymbol{K}}(\boldsymbol{Y} - \boldsymbol{H}\boldsymbol{X}^b) \\
\hat{\boldsymbol{K}} = \hat{\boldsymbol{P}}^{\boldsymbol{b}}\boldsymbol{H}^T(\boldsymbol{H}\hat{\boldsymbol{P}}^{\boldsymbol{b}}\boldsymbol{H}^T + \boldsymbol{R})^{-1}
\end{cases}
\tag{3}
$$

where $\boldsymbol{X}^a \in \mathbb{R}^{n \times N}$ is the analysis ensemble, $\boldsymbol{K} \in \mathbb{R}^{m \times n}$ the Kalman gain, $\boldsymbol{H}$ the observation operator that maps the background to the observation space, and $\boldsymbol{R} \in \mathbb{R}^{m \times m}$ the observation error covariance. The observation covariance matrix $\boldsymbol{R}$ is

approximated by its ensemble representation $\boldsymbol{R_e}$, defined as $\boldsymbol{R_e} = \overline{\boldsymbol{\epsilon}\boldsymbol{\epsilon}}^T$ (Evensen, 2003).

## 3.2   Observation errors

For both daily DA and yearly DA, correlations between observation errors in space are neglected. This assumption is strong but common in data assimilation applications, due to the lack of available information on potential correlations (Carrassi et al., 2018). In practice, this corresponds to a diagonal observation error covariance matrix $\boldsymbol{R}$, which is filled with the observation

error variance $\boldsymbol{\sigma}^2_{\mathrm{obs}}$. In order to define at best $\boldsymbol{\sigma}^2_{\mathrm{obs}}$, two different approaches have been implemented depending on the DA time scale.

### 3.2.1   Daily DA

Errors derived from the maintained performance ($\sigma_{\mathrm{MP}}$) and the site representativeness ($\sigma_{\mathrm{SR}}$) are available during the 1999-2012 period, and used to define the measurement error, assuming that the two types of errors are Gaussian:

$\sigma_{\mathrm{obs}} = \sqrt{\sigma^2_{\mathrm{MP}} + \sigma^2_{\mathrm{SR}}}$ (4)

Before 1999, only the type of station is available (see Sect. 2.3), and it is used to provide an estimate of observation error as in Devers et al. (2020a). Type 0 and type 1 stations are classified as class B for the maintained performance and class 2 for the representativeness of the site (see Table 1a and 1b). Stations with a type higher than 1 are classified as class C for the maintained performance and class 3 for the representativeness of the site. For precipitation, the minimum standard deviation is

set at 1 mm. Equation 4 is used here again to derive the estimated measurement error.

### 3.2.2   Yearly DA

No metadata on the quality of observations aggregated at yearly time scale is available. However, the work of Moisselin et al. (2002) and Gibelin et al. (2014), and a graphical analysis of long-term stations allow to provide a rough estimate of $\boldsymbol{\sigma}^2_{\mathrm{obs}}$ at the yearly time scale :

$\boldsymbol{\sigma}^2_{\mathrm{obs}}[T_{\mathrm{obs}}] = 0.5°C$

(5)

$\boldsymbol{\sigma}^2_{\mathrm{obs}}[P_{\mathrm{obs}}] = 20\% \times P_{\mathrm{obs}}$





with $T_{\text{obs}}$ the yearly-average observed temperature and $P_{\text{obs}}$ the yearly-accumulated observed precipitation. The observation error as defined here is purposely on the upper range for both temperature and precipitation to take into account the lack of information at this time scale.

### 3.3 Observation operator

The observation operator $\boldsymbol{H}$ was validated in Devers et al. (2020a) on the 1999-2012 period. $\boldsymbol{H}$ is linear and identical for both daily and yearly DA but varies slightly according to the variable considered.

At each time step $t$, an altitudinal gradient $\alpha(t)$ is computed using the background values in a linear regression. $\alpha$ is estimated within each climatologically homogeneous zone (Fig. 1). Moreover, if the altitude difference between the cells is greater than 300 m, the zone is again split by bandwidth of 300 m. At each time step, the following formula is thus applied:

$$\boldsymbol{H}\boldsymbol{X}^b(t) = \boldsymbol{X}^b(t) + \alpha(t) \times (Alt_{\text{cell}} - Alt_{\text{station}}) \tag{6}$$

with $\alpha(t)$ a vector containing the altitude gradient by zone defined previously, $Alt_{\text{cell}}$ the altitude of the cell, $Alt_{\text{station}}$ the altitude of the measurement station and $t$ the time index.

For precipitation, in order to limit the noise due to small altitude differences, $\alpha(t) = 1 \; \forall t$ when $Alt_{\text{station}} - Alt_{\text{cell}} \leq 300\,m$, and when all background member values are null.

### 215 3.4 Localization matrices

Considering the 25-member ensemble size of the background, a localization is applied on the background error covariance matrix to reduce or even remove covariances that seem physically erroneous (Houtekamer and Mitchell, 1998; Houtekamer and Zhang, 2016). Equation 3 of the Kalman gain becomes:

$$\hat{\boldsymbol{K}} = [\rho \circ (\hat{\boldsymbol{P}}^b \boldsymbol{H}^T)][\rho \circ (\boldsymbol{H}\hat{\boldsymbol{P}}^b \boldsymbol{H}^T) + \boldsymbol{R}]^{-1} \tag{7}$$

with $\rho \in \mathbb{R}^{m \times m}$ the localization matrix and $\circ$ an element-wise (Schur) product.

The localization matrix is generally built on a specific distance representative of the decorrelation distance inside the variable (Anderson, 2012). However, these approaches rely on the assumption than the error is isotropic. This assumption may be wrong with respect to daily precipitation and temperature at high resolution (8 km). Hence, the localization matrices $\rho$ are here built upon the background climatology in such a way that a plausible anisotropic behaviour is intrinsically integrated (see Devers
et al., 2020a).

The correlation matrices are computed as follows :

  – For the daily DA, the seasonally-adjusted daily time series of SCOPE Climate over the 1958-2008 period are extracted. The Pearson correlation coefficient between each pair of cells is then computed for each member, leading to 25 correlations matrices.

– For the yearly DA, the yearly time series of SCOPE Climate over the 1958-2008 period are used. Once again the Pearson correlation is computed for each of the 25 members.





The correlation matrices are then processed in the same way for both the daily and yearly DA. First, a matrix $\rho_1$ is computed as the element-wise median of the 25 correlation matrices previously created. Inside a given climatically homogeneous zone, correlations are close to 1, resulting from the hypothesis made originally in Safran and transferred to SCOPE Climate. To re-

move this strong hypothesis of climatologically homogeneous zones, a second correlation matrix $\rho_2$ is based on an exponential function of the distance between cells. This function is calibrated for each cell allowing to have a larger radius in areas with oceanic climate and a smaller one in mountains for example (for more details, see Devers et al., 2020a). An element-wise product of the two matrices allows to obtain the final localization matrix $\rho$ :

$$\rho = \rho_1 \circ \rho_2 \tag{8}$$

Localization matrices $\rho$ thus hold an anisotropic behaviour and allow different values inside the climatologically homogeneous zones (Figure 2).

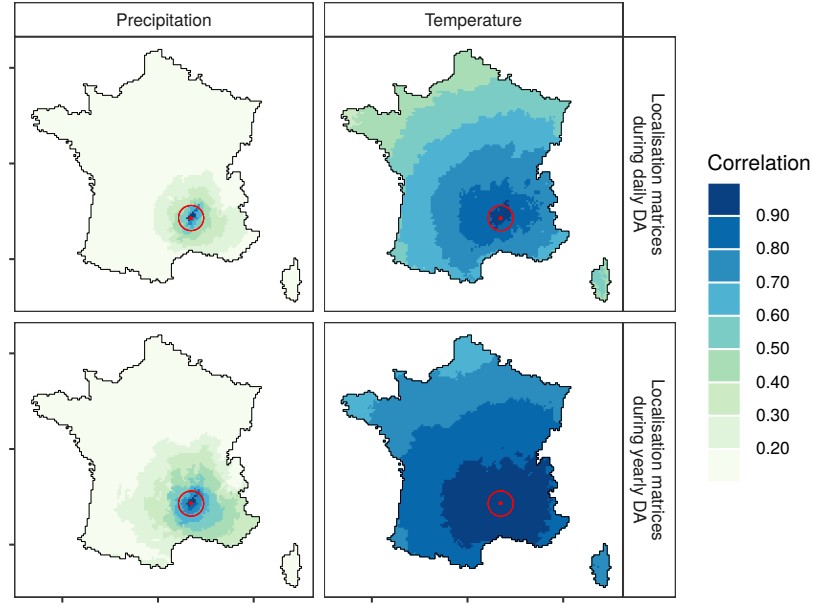

**Figure 2.** Correlation matrices between values from the case study cell (in red) and all other cells, for daily (upper panels) and yearly (lower panels) data assimilation, for precipitation (left) and temperature (right).

### 3.5 Precipitation transformation

The Ensemble Kalman fitting scheme is optimal in a Gaussian framework, but daily and yearly precipitation follows a positive, skewed, and asymmetric distribution with a spike at zero for daily precipitation (Figure 3). However, the non-normality of

daily precipitation is often neglected in data assimilation (e.g., Quintana-Segui et al., 2008; Bhargava and Danard, 1994; Soci



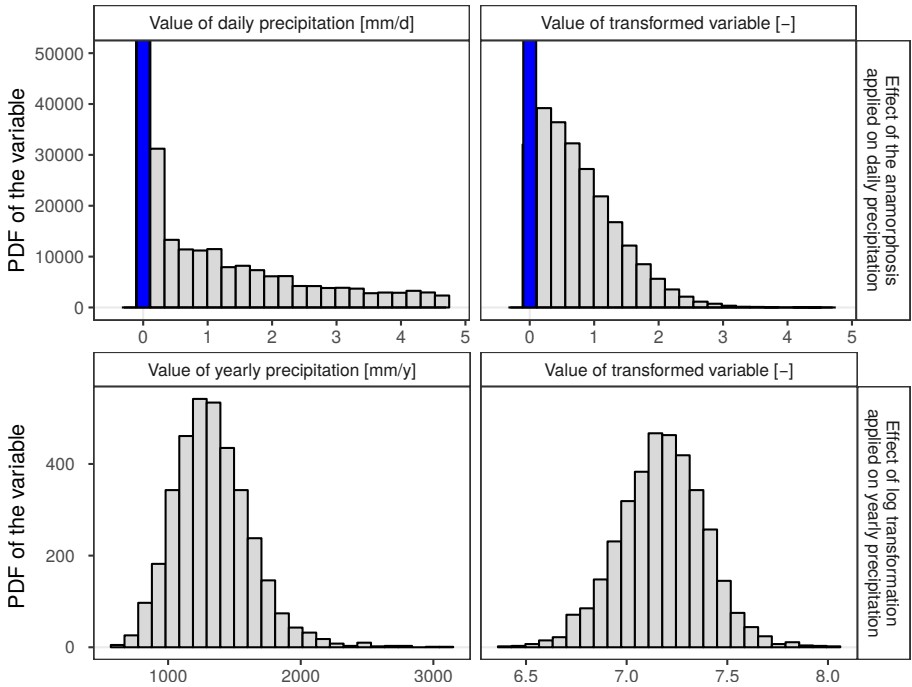

**Figure 3.** Schematic view of the transformations applied to SCOPE Climate daily (upper panels) and yearly (lower panels) precipitation for the case study cell during the 1958–2008 period. The blue line represents zero values. For daily precipitation the x–axis of left panels is truncated at 5 mm/d for readability.

et al., 2016), while Mahfouf et al. (2007) assume a lognormal distribution. Lien et al. (2013) and Devers et al. (2020a) applied an anamorphosis to precipitation, that consists in projecting the daily precipitation into a normal space where the analysis is carried out, and mapping the analysis back into the original space using the inverse of the transformation (Wackernagel, 2003; Bertino et al., 2003). Devers et al. (2020a) showed that the impact of the Gaussian anamorphosis on daily precipitation is
lower than the impact of localization, but that it improves estimates in areas with sparse observations. In the current study, two different strategies are selected to transform the precipitation depending on the DA time scale.

### 3.5.1   Daily DA

An anamorphosis transforming the raw daily precipitation $X$ into a transformed variable $Z$ is applied as follows:

$$Z = G^{-1}[F(X)] = \sqrt{2} \times erf^{-1}(2 - F(X)) \tag{9}$$

with $F(X)$ the cumulative density function $X$, $G$ the cumulative density function of $Z$, and $erf^{-1}$ the inverse error function.
    The anamorphosis is defined locally for each grid cell with $X$ the ensemble from SCOPE Climate during the 1958-2008 period, and the function is then piecewise-linearized (Simon and Bertino, 2009; Brankart et al., 2012). Outside of this period, the following rules are applied. Considering $X_{\min}$ and $X_{\max}$ the limit of the function domain, if $X_{\text{zero}} < X < X_{\min}$ then $Z[X] =$





$G^{-1}[F(X_{\text{zero}})]$. If $X > X_{\max}$, then a linear regression fitted on values higher than the 99th percentile of non-zero precipitation
is used, meaning that the tail of the transformed distribution is considered as gaussian (Devers et al., 2020a). However, even
with the anamorphosis, the distribution obtained is closer to a truncated gaussian pdf than a true gaussian pdf (see Figure 3,
top panel).

### 3.5.2 Yearly DA

For yearly precipitation, a simpler approach is implemented, assuming that yearly precipitation follows a log-normal dis-
tribution for each cell, thus making extrapolation more straightforward (Figure 3). Yearly precipitation values $X$ are thus
transformed as follows, adding a 1 mm offset to allow for transforming null precipitation :

$$Z = log(X+1) \tag{10}$$

### 3.5.3 Common processing

Irrespective of the time scale, the above transformation functions are applied before the analysis on (1) the background values,
(2) the observations, and (3) the standard deviations. For the standard deviations, the non-linearity of the transformations is
taken into account as follows (see Lien et al., 2013):

$$\sigma_{\text{trans}} = \frac{[(y+\sigma)_{\text{trans}} - y_{\text{trans}}] + [y_{\text{trans}} - (y-\sigma)_{\text{trans}}]}{2} \tag{11}$$

with $y$ the observation vector in the original space, $\sigma$ the associated error, and the index $_{\text{trans}}$ indicates the variable transformed
in the Gaussian space. After the analysis step, the analysis state $\boldsymbol{X}^a$ is then transformed back into the original space with the
reciprocal functions of the anamorphosis and the logarithmic transformation.

### 3.6 Production of the reanalyses over 1871-2012

This section describes how the different reanalyses are produced over the 1871-2012 period (Figure. 4).

### 3.6.1 Application of the Ensemble Kalman fitting

The EnKf described in Sect. 3.1 is here applied for the two time scales. The FYRE Daily reanalysis is created using the scheme
proposed by Devers et al. (2020a). The assimilation is done independently each day from 1 January 1871 to 29 December 2012
using the 25 SCOPE Climate members of temperature and precipitation as the background. The assimilated observations are
daily in-situ measurements of temperature and precipitation originating from the Météo-France database. FYRE Daily is thus
a daily gridded reanalysis composed of 25 time series of precipitation and temperature fields.

The FYRE Yearly reanalysis is produced using yearly-averaged temperature values and yearly-accumulated precipitation.
Once again the background is given by SCOPE Climate, and observations from the Météo-France database are assimilated (see
Sect. 2.2). The assimilation is applied each year independently between 1871 and 2012, leading to the FYRE Yearly reanalysis
composed of 25 yearly-averaged gridded time series of precipitation and temperature fields.





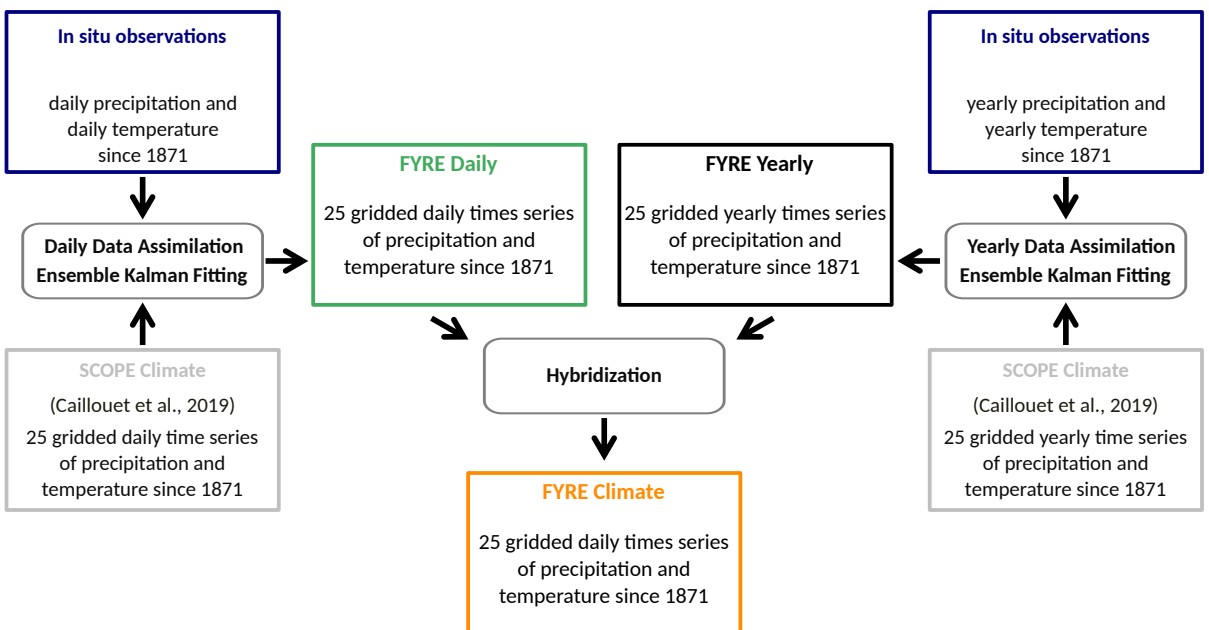

**Figure 4.** Production scheme of the different reanalyses. Details can be found in Sect. 3.6.

In FYRE Daily and FYRE Yearly, the assimilation is performed independently for temperature and precipitation, and independently at each time step. This means that assimilating precipitation has no impact on the temperature analysis (see the discussion in Devers et al., 2020a), and that assimilating an observation at a given time step has no effect on the analysis at another time step.

### 3.6.2 Hybridization

Finally, the FYRE Climate daily product combining the information of the daily and yearly reanalyses is derived through an hybridization between FYRE Daily and FYRE Yearly, following approaches adopted in numerous meteorological studies (Magand et al., 2018; Sheffield et al., 2006) and paleoclimates studies (Dirren and Hakim, 2005; Steiger and Hakim, 2016; Huntley and Hakim, 2010). The hybridization here aims at transforming daily values from FYRE Daily to match yearly values from FYRE Yearly.

For temperature, an additive transformation is commonly used and is adopted here (Dirren and Hakim, 2005; Steiger and Hakim, 2016; Huntley and Hakim, 2010). For precipitation, a multiplicative transformation is commonly used and is adopted here (Ngo-Duc et al., 2005; Keller et al., 2015). Note that such a transformation leads to largest changes in higher precipitation values, and that dry days from FYRE Daily will remain unchanged in FYRE Climate.





Each member of FYRE Climate is generated as follows. First, the ratio of precipitation $\beta$ and temperature $\alpha$ are computed for each year based on the annual values of FYRE Yearly and FYRE Daily:

$$\boldsymbol{\beta}[y,c] = \frac{P_{\text{yearly}}[y,c]}{P_{\text{daily}}[y,c]}$$

$$\boldsymbol{\alpha}[y,c] = T_{\text{yearly}}[y,c] - T_{\text{daily}}[y,c] \tag{12}$$

where $y$ and $c$ are the year and cell considered, $P$ and $T$ the value of precipitation and temperature, respectively, and the index defines the dataset considered: *daily* for FYRE Daily and *yearly* for FYRE Yearly. Then, the times series of FYRE Climate are computed using the previously defined ratio and the daily time series of FYRE Daily:

$$P_{\text{climate}}[d,c] = P_{\text{daily}}[d,c] \times \boldsymbol{\beta}[y,c]$$

$$T_{\text{climate}}[d,c] = T_{\text{daily}}[d,c] + \boldsymbol{\alpha}[y,c] \tag{13}$$

with notations as above, and $d$ the day considered. The *climate* index refers to the final FYRE Climate values. This process leads
to two daily 25-member ensemble products over the 1871-2012 period: FYRE Daily and FYRE Climate, whose differences are assessed below.

## 4 Results

A first part of the results section is dedicated to the comparison between SCOPE Climate/FYRE Daily/FYRE Climate, and (1) the Safran reanalysis, (2) the monthly homogenized series (SMR) and (3) the European Pattern Climatology (EPC). A second
part will provide examples of time series and extreme events to give a more precise idea of the characteristics of each dataset.

### 4.1 Comparison with the Safran reanalysis

For temperature, over the 1960-2000 period, the behaviour of FYRE Daily and FYRE Climate is similar to the Safran reanalysis with a low CRPS and a high daily correlation (see Fig. 5, left panels). The impact of DA can be evaluated by comparing the background and reanalysis metrics. SCOPE Climate shows a higher CRPS and a lower daily correlation with Safran, but a
slightly lower daily bias than the two reanalyses for specific areas. These differences can be explained by the assimilation of mean daily temperature that is computed using the minimum and maximum temperature, while the mean daily temperature in Safran is computed from hourly data. Biases shown by FYRE Daily are highly reduced in FYRE Climate, showing the benefits of the hybridization.

The right panels of Fig. 5 demonstrate the interest of DA concerning precipitation. The FYRE Daily and FYRE Climate
reanalyses have a much lower CRPS and a much higher correlation with Safran than SCOPE Climate all over France. Although some differences are of opposite sign on contiguous cells, there is a clear underestimation of FYRE Daily precipitation in mountainous areas, which is highly reduced in FYRE Climate, reaching values between -5% and 5%.

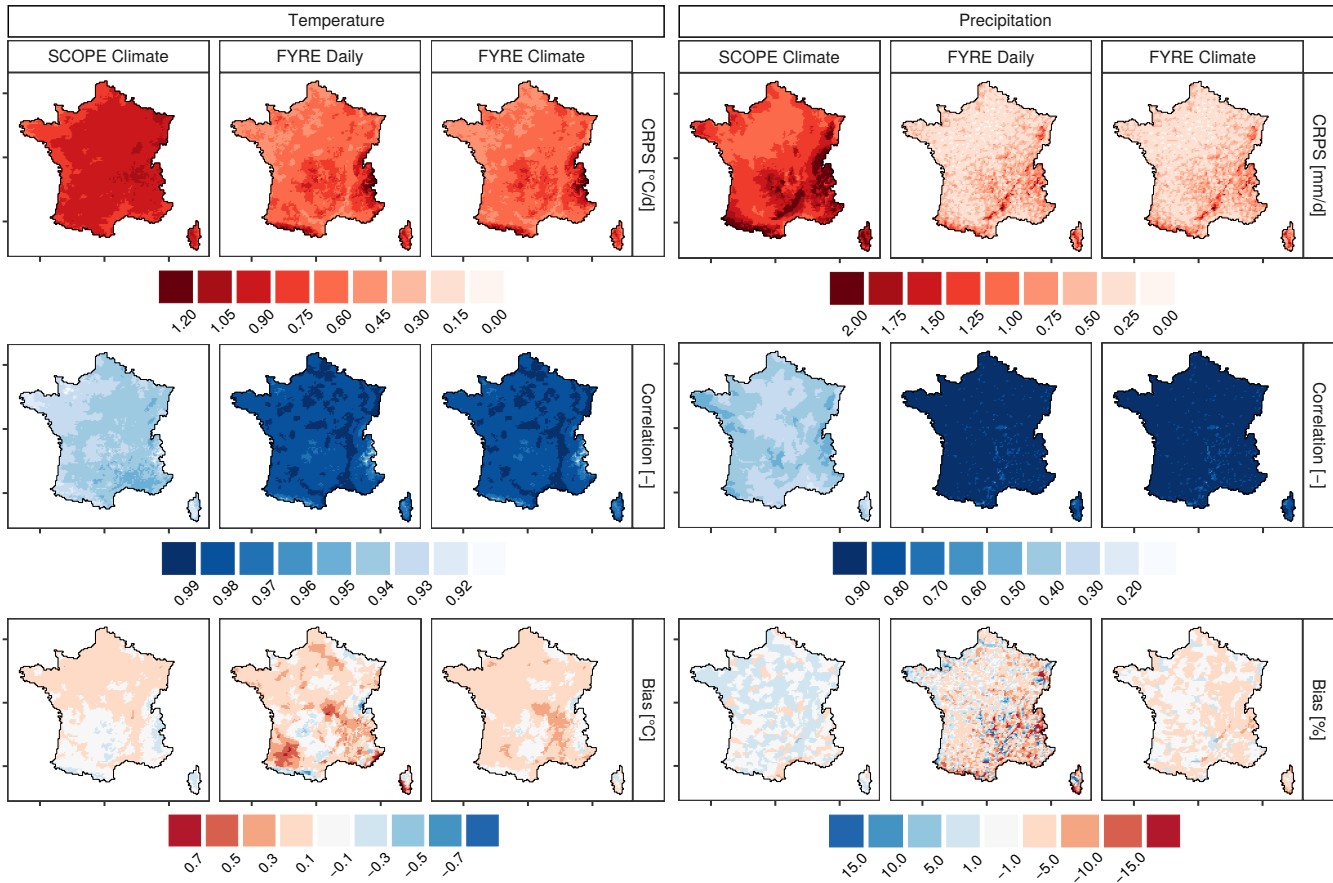

**Figure 5.** Mean of daily Continuous Ranked Probabiliy Score (CRPS) (top row), daily correlation (middle row) and daily bias (bottom row) between Safran and SCOPE Climate/FYRE Daily/FYRE Climate for the 1960-2000 period for temperature (left panels) and precipitation (right panels). Correlation and bias are median values over each ensemble.

## 4.2 Comparison to the the monthly homogenized series

In order to produce a verification constant over time – i.e with a rather steady number of validation stations – the analysis is
divided in two periods. The reanalyses are compared to the monthly homogenized series (SMR) over the 1959-2009 period and the 1900-2000 period, that include 1583 and 332 stations respectively for precipitation and 308 and 88 for temperature (see Sect. 2.4.2). Scores (bias, correlation, and RMSE) are computed for each station and then averaged over France to provide a synthetic assessment of the performance with respect to SMR.

### 4.2.1 Over the 1959-2009 period

For temperature, the Safran reanalysis is negatively biased with respect to SMR (Fig. 6, left panels). This difference is probably induced by differences in the computation of the mean daily temperature (see above), and the non-stationarity of the bias over



time could reflect the asymmetric evolution of the minimum and maximum temperature. The bias of the background SCOPE Climate is around 0 at the start of the period and slowly degrades towards negative values, resulting from an underestimation of the recent warming already noted by Caillouet et al. (2019). FYRE Daily and FYRE Climate both display a much smaller

negative bias – with values around -0.2°C – and relatively constant over the last 30 years. SCOPE Climate has a lower correlation than all other products over the entire period. The FYRE Daily and FYRE Climate reanalyses show a higher correlation than Safran, and an uncertainty – defined by the spread of the ensemble – quite reduced compared to SCOPE Climate. A similar analysis may be drawn for the RMSE. FYRE Daily shows slightly lower RMSE values than FYRE Climate, but the two reanalyses perform overall similarly, and much better than SCOPE Climate or even Safran.

For precipitation, SCOPE climate shows a bias with a high interannual variability (Figure 6, top-right panel). All reanalyses including Safran show a very low and constant bias, with a very small spread for FYRE Daily and FYRE Climate. The impact of DA on correlation is also very clear, with a 0.3 increase in average for FYRE Daily and FYRE Climate compared to SCOPE Climate. Once again, the spread of the ensemble is reduced through the DA over the entire period. Finally, the Safran reanalysis has slightly lower correlations than the FYRE Daily and the FYRE Climate reanalyses. The RMSE is four times

higher in SCOPE Climate than in the reanalyses. Among those, FYRE Daily shows the lowest errors, followed by FYRE Climate and then by Safran.

Figure 6 shows an overall large impact of the DA that allows FYRE Daily and FYRE Yearly to reach higher performances (lower bias, higher correlation, lower RMSE) than Safran – the current reference reanalysis over the 1959-2009 period – compared to the monthly homogenized series.

### 4.2.2    Over the 1900-2000 period

Most of the comments made above with the most recent SMR dataset are also valid here for the post-1950s period, and a focus is thus made on centennial evolutions.

The average bias of temperature between SMR and SCOPE Climate roughly varies between -0.5°C and +0.5°C (Fig. 7, top-left panel). Before the 1950s, the two reanalyses do not share the same bias characteristics: FYRE Daily shows a slightly

positive bias as well as a strong reduction of the ensemble spread after the 1900s, while FYRE Climate shows a strong dependency to the background and an ensemble spread which gradually shrinks over the 20th century. The correlation of the two reanalyses with SMR is clearly linked to the density of assimilated stations (see Fig. 1), with slightly reduced values before 1950 and drops during the two world wars. Nevertheless, values are constantly higher than those of the background. When RMSE for SCOPE Climate do not show any trend over the 20th century, those of the two reanalyses show a steady decrease

from 0.7°/month in 1900 to 0.4°in 2000, only interrupted during the second world war as a consequence again of the drop in assimilated observations.

For precipitation, the background shows a global overestimation during the 1900-1960 period and an overall bias close to zero afterwards, but with a high interannual variability (+30% to -10%) (Fig. 7, top-right panel). The absolute bias values are rather constant and much lower for the two reanalyses, albeit slightly increasing towards the beginning of the century. FYRE

Daily (resp. FYRE Climate) shows a slightly negative (resp. positive) bias before 1960. FYRE Daily also shows an intriguing





split of the ensemble before 1960, which will be discussed in Sect. 4.4 below. As for temperature, the correlation of the two reanalyses is quite higher than those of the background, with slightly lower values during the first half of the century. The RMSE pattern is similar to that of temperature, with a steady decrease for the two reanalyses over the century, ranging from 20 mm/month in 1900 to around 10 mm/month in 2000, when SCOPE Climate values vary around 40 mm/month.

Overall, and beyond the evolution of ensemble-average values, the spread of the two reanalyses tend to shrink over the course of the century for all indicators, following the increasing number of assimilated observations.

## 4.3 Multidecadal variability

The long-term consistency between different datasets allows to further evaluate the two reanalyses. To that end, anomalies are computed over the 1871-2012 period for several long-term datasets described in Sect. 2.4, using the 1900-2000 period as
a reference. Anomalies are computed for each cell over France (see Sect. 2.4 for the number of cells in each dataset). Their median value is retained, smoothed with a 20-year rolling mean, and plotted in Fig. 8. Smoothed anomalies are computed for each member when available. EPC and SMR show a similar evolution of both temperature and precipitation over the 20th century. Negative temperature anomalies are found before the 1940s – from around -0.35°in 1910 and down to -0.6°in 1880 for EPC – and around the 1970s, and positive ones for other periods, with a steep recent warming from the 1980s onward reaching
0.5°in 1990 (Fig. 8, top-left panel). Negative precipitation anomalies are found before 1910 and around the 1940s-1950s, and positive ones in other periods, including the most recent one.

For temperature, SCOPE Climate anomalies are rather consistent with those from EPC and SMR over the 20th century. However, SCOPE Climate shows much higher – but still negative – anomalies than EPC before 1900, and underestimate the recent warming compared to EPC and SMR. FYRE Daily anomalies are closer to those of EPC and SMR after 1940 compared
to SCOPE Climate – including during the recent warming –, but the original discrepancy at the beginning of the period extends to 1940. FYRE Climate anomalies are quite consistent with those of EPC and SMR from 1910 onward. However, before 1910, they are roughly constant around -0.2°, i.e. much less negative than those of EPC. This discrepancy may come from the non-homogeneity in underlying data in EPC: gridded observations after 1900 and climate field regression before that (see Sect. 2.4).

For precipitation, the high multidecadal variability of SCOPE Climate leads to positive anomalies over the 1890-1930 period, with values nearly reaching +10%, when EPC and SMR values are only slightly positive. This is probably a bias inherited from the 20CR driving global extended reanalysis. Indeed, Bonnet et al. (2017) found much higher positive anomalies in 20CR precipitation over France than in the SMR over this period. SCOPE Climate also shows negative anomalies from 1960 onward when both EPC and SMR show positive anomalies. The overall multidecadal evolution of FYRE Daily is much more
consistent with those of EPC and SMR, and the ensemble spread is quite reduced with respect to the background SCOPE Climate. However, anomalies are systematically shifted towards lower ones by 2 to 3% before 1950 and to higher ones – up to +5% – after 1970, showing that assimilating daily observations only does not allow to accurately reproduce the multidecadal variability. Lastly, FYRE Climate is much more consistent to EPC and SMR long-term evolution, even with a small spread, as small as that of FYRE Daily.





## 4.4 Time series analysis

Time series over the Cévennes case study cell (see Fig. 1) derived from observations, Safran, SCOPE Climate, FYRE Daily, and FYRE Climate are presented in Fig. 9 and Fig. 10 to exemplify the behaviour of the different datasets at different time scales and for selected periods differing in the amount of data assimilated: 1871-2012 at the annual time scale, years 1900, 1936, and 2000 at the monthly time scale, and June 1900, June 1936, and June 2000 at the daily time scale.

For temperature, all long-term datasets are well correlated at the annual time scale, but FYRE Daily values are systematically hotter before 1950. The ensemble spread is rather constant for SCOPE Climate while it is shrinking in the two reanalyses when more observations are assimilated. The underestimation of the recent warming by SCOPE Climate is once again visible here. The amplitude of the annual cycle for the three years considered appears underestimated in SCOPE Climate compared to the two reanalyses, an issue already identified by Caillouet et al. (2019) with respect to Safran. At the daily time scale, the ensemble spread is much reduced in both reanalyses compared to SCOPE Climate, even more so for June 2000 when many observations are assimilated close to – and not within – the case study cell considered.

For precipitation, Fig. 10 shows that DA tends to reduce the ensemble spread at the annual time scale even at the beginning of the period when only few data are assimilated. Large discrepancies are found for specific years between SCOPE Climate, FYRE Daily and FYRE Climate. More specifically, extreme values are found for FYRE Daily in e.g. 1879 and 1936, the latter year also showing a split of the ensemble, already noted earlier in Fig. 7. Similar comments may be drawn at the monthly and daily time scales, including the ensemble split over 1936, which is also present in FYRE Climate, but to a lesser extent. The puzzling behaviour of FYRE Daily is in fact explained by two stations located close to the cell – less than 10km, hence with high covariances – that give contradictory input to the DA scheme. Indeed, the two stations – #7235003, Sainte-Eulalie, 1350m a.s.l., and #7326003, Usclades-et-Rieutord, 1270m a.s.l. – both start providing precipitation data on 1 January 1936 and are assimilated, but with very different daily amounts (not shown here). At the end of 1936, the station #7235003 is closed, and FYRE Daily then shows a much more coherent ensemble as seen at the yearly time scale (Fig. 10, top panel). It is interesting to note that the hybridization leads to a much reduced ensemble split, showing an unexpected advantage of FYRE Climate.

## 4.5 Examples of extreme events

The impact of DA on the representation of extreme events is here investigated on two events: (1) the cold month of December 1879 over the North-East of France (Figure 11), and (2) an extreme precipitation event in the Cévennes area on 21 September 1890 (Figure 12). To that end, three members – #8, #15 and #19 – have been randomly selected from SCOPE Climate, FYRE Daily and FYRE Climate and are compared with the available observations at the time.

### 4.5.1 An extreme cold wave

December 1879 is an extremely cold month in France as shown by the frost of the Loire, the Seine, the Saône and the Rhône rivers (Dubrion, 2008) with a negative anomaly of -10.2°C (Le Roy Ladurie et al., 2011, p. 202). The Annals of the Central Meteorological Office of France describe in details the anticyclonic state lasting most of the month and the consequent very





cold temperature over France and central Europe (Angot, 1881, p. 19-23). Minimum values dropped for example below -25°in Paris on 10 December (Le Roy Ladurie and Séchet, 2009, p. 43-44). The specificity of this cold wave is its duration, which led to December-averaged daily mean temperature reaching values well below -5°C in the North-East of France, and even -10.3°C

for the Commercy station (id: 55122003). In order to obtain a more detailed validation of this event, monthly independent observations have been digitized from the Annals of the Central Meteorological Office of France (Mascart, 1881, p. 217-240).

Figure 11 shows the December-averaged temperature over France, in the assimilated observations, in the independent observations, in the background SCOPE Climate, and in the two reanalyses. Compared to the observations, SCOPE Climate members largely overestimate the temperature everywhere except around the Mediterranean. This is especially true in the

North-East, with more than 3°C discrepancies. The impact of DA is quite clear, with both reanalyses showing much colder values, thanks to only 18 unevenly distributed assimilated stations. FYRE Climate is slightly less cold than FYRE Daily, but differences are overall minor. The independent observations confirm both the location and the intensity of the extreme cold temperature given by the two reanalyses, with e.g. -9.4°C in Troyes and -9.55°C in Mirecourt located in the center of the event and a larger area with temperatures between -7 and -9°C. The independent stations located in the South and West of France

also allow to grasp the positive impact of the DA outside the area impacted by the cold event.

### 4.5.2 An extreme precipitation event

At the end of September 1890, an extreme rainfall event in the Cévennes area[1] led to a record flood over the Ardèche river between the 21 and 23 September 1890 (Sheffer et al., 2003; Naulet et al., 2005). Extreme precipitation amounts were recorded from 18 to 23 September reaching 971 mm at the Montpezat station (Météo-France, 1995, p. 26-27). Figure 12 focuses on 21

September, when the highest daily amount of precipitation – 346 mm at Saint-André de Valeborgne, id: 30231001 – was recorded, with similar very high values in a small area oriented South-West to North-East (Fig. 12, left panel). Observations are mainly located in the central part of the Cévennes area, with few or no station further north or south, thus impeding a global view of the event. The first two selected members of SCOPE Climate display very low precipitation values compared to observations, when the third one reaches values higher than 250 mm, but still underestimating recorded values. This latter

member furthermore provides a spatial pattern of precipitation consistent with the classical shape of heavy precipitation events – called Cévenol events – in this region (see e.g. Boudevillain et al., 2016). This high uncertainty in SCOPE Climate is dramatically reduced through DA, with both reanalyses providing precipitation values much closer to the observations, with amounts reaching 400 mm – i.e. exceeding recorded ones – in some cells. A similar spatial pattern of the event is given by the two reanalyses, with a north-eastern extension. Remaining differences between members reflect the uncertainty due the lack

of observations, notably to the north-east, where reanalyses still suggest very high values. This example shows that DA thus allow to strongly reduce the uncertainty and to produce gridded meteorological fields more coherent with in-situ observations.

---

[1]For an extended description of the event, see http://pluiesextremes.meteo.fr/france-metropole/Inondations-en-Cevennes-Crue-historique-de-l-Ardeche.html





## 5 Discussion

### 5.1 Transforming precipitation

The anamorphosis chosen for transforming daily precipitation had already been applied with a large improvement of the
analysis by Lien et al. (2013), and with a smaller one by Devers et al. (2020a). Implementing the anamorphosis however
requires additional choices for extrapolating values, to both very low and positive values and to very high values (Lien et al.,
2013, 2016). Choices are here made following Devers et al. (2020a).

A logarithmic transformation is here applied to yearly precipitation. The impact of this transformation has been studied in
an experimental set-up similar to the one proposed by Devers et al. (2020a) for daily precipitation, by varying the density of
assimilated stations over the 1950-200 period and evaluating the analysis on independent data. These experiments showed that
the logarithmic transformation allow for cancelling a dry bias in the analysis resulting from DA without transformation (not
shown).

### 5.2 Estimating the observation error

Applying the DA scheme over the 1871-2012 period has put forward the need for quantifying the observation error, a key
variable in the analysis step.

For the daily DA, the 1999-2012 period is rich in metadata, allowing for precisely defining the observation error based
on the work of Météo-France and the World Meteorological Organization (WMO, 2014). Before 1999, the type of station is
the only relevant metadata available. Devers et al. (2020a) translated the type of station into the framework of measurement
errors linked to the maintained performance and site representativeness (Leroy, 2010; Leroy and Lèches, 2014) and found that
making such an hypothesis improved both the reanalysis uncertainty and its reliability. This approach thus makes the most of
available information, by distinguishing two classes of stations and associated measurement errors when no other metadata are
available.

Estimating the observation error is even more difficult for the yearly DA, as no information is available at this time scale.
Estimates used here may seem large (see Sect. 3.2), but a conservative choice has been made here to reflect e.g. the homogeneity
breaks than can be observed in the annual temperature and precipitation times series for some long-term stations. Further
investigation on the yearly estimates of the observation error could focus on the intensity of the correction applied during the
homogenization process of the SMR (Moisselin et al., 2002; Gibelin et al., 2014) at the yearly time step.

### 5.3 On the background uncertainty

The background for DA – SCOPE Climate – comes from a downscaling of the ensemble-mean fields of the 56-member Twen-
tieth Century Reanalysis (Compo et al., 2011). SCOPE Climate may therefore underestimate the reconstruction uncertainty,
especially at the end of the 19th century when few pressure observations were available, as discussed by Caillouet et al.
(2016). Caillouet et al. (2019) showed that SCOPE Climate ensemble spread is presumably too small for temperature at Paris-



Montsouris station. However, DA experiments made here show that the background uncertainty is yet large enough to led to very satisfactory results even before 1900 (see Sect. 4.5).

## 5.4 Assimilating yearly-averaged observations

The assimilation of yearly observations allowed to recover multidecadal variations consistent with other products such as SMR and EPC, as opposed to daily-only DA. For precipitation, this gain could be linked to the non-Gaussian properties of daily precipitation – even after anamorphosis – in contrary to log-transformed yearly precipitation. However, it is not so clear why this is the case of daily temperature. Nonetheless, Steiger and Hakim (2016) showed that assimilating low-frequency data improves the low-frequency components of reconstructions compared to using high-frequency data only.

In addition, most of the observations are assimilated twice in FYRE Climate: through the daily DA, and through the yearly DA. This can be problematic, as it could lead to an overconfident reanalysis. However, in this case and similarly to recent paleoclimate DA studies (Steiger and Hakim, 2016, see e.g.), when the background is composed of the same dataset for the two time scales, the daily reanalysis is not used as a background for the yearly DA, thus maintaining relative independence.

## 5.5 On the hybridization

Choices made for the hybridization build on previous studies, notably for the additive formulation for temperature, in paleoclimate after DA of time-average observations (Steiger et al., 2014; Steiger and Hakim, 2016; Dirren and Hakim, 2005; Huntley and Hakim, 2010), but also for the more recent climate (Ngo-Duc et al., 2005; Weedon et al., 2011; Sheffield et al., 2004). A step further would be to make these additive corrections to also depend on the season. To that end, the most direct approach would be to assimilate temperature observations at the monthly or seasonal time scales.

The multiplicative correction applied to precipitation has been implemented following the work of Ngo-Duc et al. (2005) and Keller et al. (2015). The intensity of the correction is thus by construction higher during the wet season than during the dry season. Moreover, the correction has an impact only on wet days and does not affect the number of dry days. Methods have been developed to modify the number of wet days (Weedon et al., 2011; Sheffield et al., 2004) using wet–wet and dry–dry conditional probabilities of the compared dataset, but Sheffield et al. (2004) also note that they may compromise the spatial consistency.

## 6 Conclusions

The present study goal was to build on the work of Devers et al. (2020a) to provide a long-term daily reanalysis of precipitation and temperature at high resolution over France. Two reanalyses were produced based on DA using the SCOPE Climate downscaled reconstruction (Caillouet et al., 2019) as background. FYRE Daily (resp. FYRE Yearly) used daily (resp. annual) observations in the DA process. These two intermediate reanalyses were then hybridized to derived the final FYRE Climate reanalysis corresponding to the study objective.

Section 4.1 showed that both FYRE Daily and FYRE Climate have strong similarities with the current reference Safran reanalysis over the period 1950-2000, and clearly improve on the SCOPE Climate background. Devers et al. (2020a) even found that FYRE Daily performs better than Safran with respect to independent data on a set of experiments over the 2009-2012 period. Section 4.2 also showed a better performance (bias, correlation, RMSE) than SCOPE Climate but also than Safran when compared to monthly homogenized time series. Section 4.5 lastly showed that both reanalyses perform very well in reproducing extreme temperature and precipitation events, which was a weak point in SCOPE Climate (Caillouet et al., 2019). All these elements clearly show the benefit of data assimilation for century-long reconstructions.

Section 4.3 highlighted the most important difference between FYRE Daily and FYRE Climate. When FYRE Daily clearly improves on the reconstruction of multidecadal variability as inferred from the SMR and EPC long-term datasets at the scale of France, FYRE Climate display variations much more consistent with these datasets than FYRE Daily, for both precipitation and temperature, and for all subperiods. FYRE Climate thus provides the best features, by performing as well as FYRE Daily on small time scales, and much better at longer time scales.

FYRE Climate is therefore the final product of this work: a daily surface reanalysis of precipitation and temperature at the daily time scale and at a 8 km resolution over France between 1 January 1871 to 29 December 2012. Moreover, FYRE Climate is an ensemble reanalysis composed of 25 members whose spread reflects the uncertainty in both the reconstruction used as background for DA, and the assimilated observations. As such, it is the first century-long surface reanalysis at a country scale, paving the way for assessing the long-term evolution of climate at the local scale and studying past extreme meteorological events. To this aim, FYRE Climate is made available freely for non-commercial purposes to the research community through two joined datasets: precipitation (Devers et al., 2020b) and temperature (Devers et al., 2020c).

*Data availability.* FYRE Climate is made available as netcdf files on the zenodo.org platform. For practical reasons, the dataset is split into one for precipitation (Devers et al., 2020b) and another one for temperature (Devers et al., 2020c). Each dataset comprises 25 netcdf files, one for each ensemble member. Please note that ensemble member #1 for temperature should be associated to member #1 for precipitation, and so on. Values are available over the Safran grid (see Vidal et al., 2010b), but only for grid cells located within France borders, as for SCOPE Climate (Caillouet et al., 2019).

*Author contributions.* AD, JPV, CL, and OV conceptualised the study. AD performed the formal analysis, conducted the investigation, developed the methodology and software with support of JPV, CL, and OV. AD wrote the original draft, prepared the visualization and JPV, CL, and OV reviewed manuscript.

*Competing interests.* The authors declare that there are no conflicts of interest regarding this work.



*Acknowledgements.* The authors would like to thank Météo-France for providing access to the Safran surface reanalysis, the monthly homogenized series, as well as to surface observations and associated metadata. Analyses were performed in R (R Core Team, 2018) with packages ncdf4 (Pierce, 2015), dplyr (Wickham et al., 2017), tidyr (Wickham and Henry, 2018), ggplot2 (Wickham, 2009), fst (Klik, 2018) and sp (Bivand et al., 2013). A. Devers PhD thesis was funded by Irstea (now INRAE) and CNR.





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

750





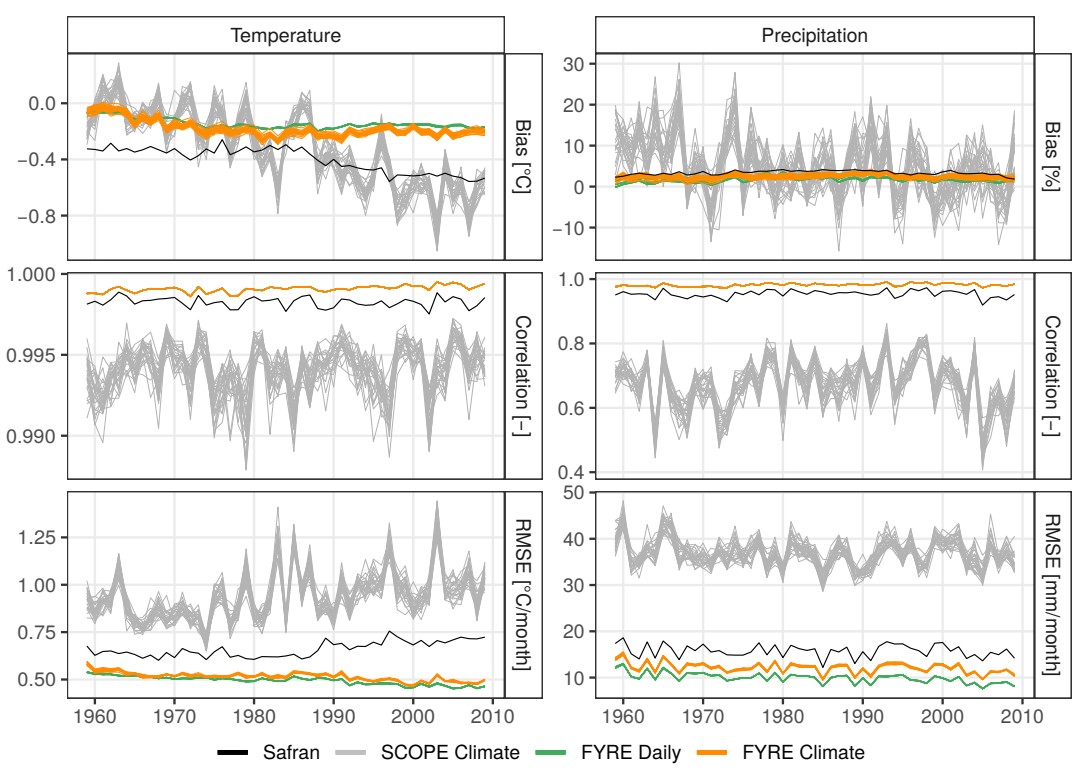

**Figure 6.** Evolution of the mean monthly bias (top row), correlation (middle row) and root mean square error (RMSE, bottom row) between the monthly homogenized series and each member of SCOPE Climate/FYRE Daily/FYRE Climate/Safran for temperature (left panels) and precipitation (right panels) between 1959 and 2009. See text for details.





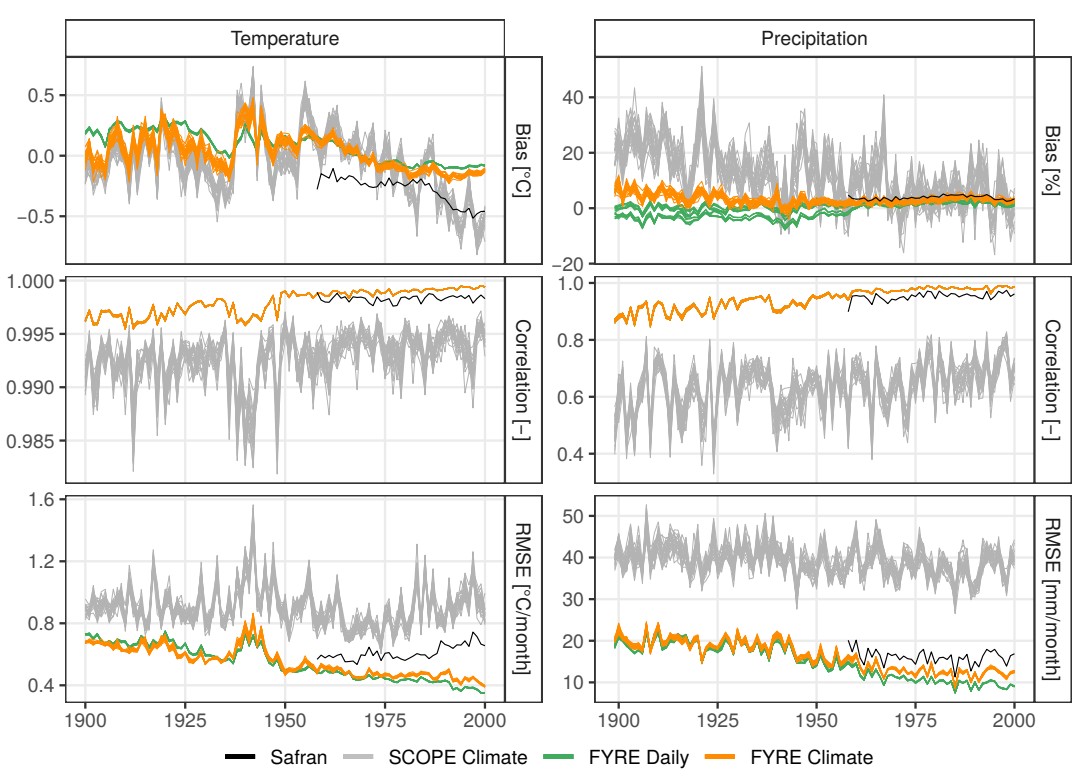

**Figure 7.** As for Fig. 6, but over the 1900-2000 period with corresponding centennial homogenized time series.

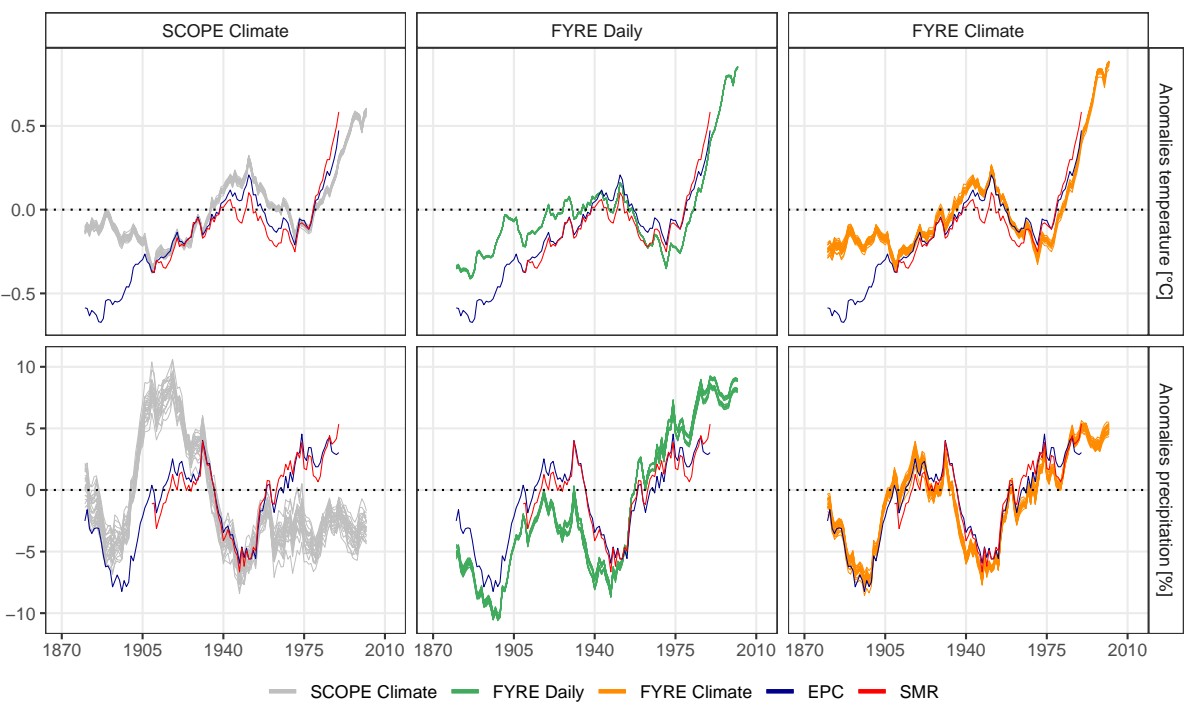

**Figure 8.** Anomalies of annual temperature (top panels) and precipitation (bottom panels) averaged over France, and smoothed with a 20-year rolling mean. See text for details.





**Figure 9.** Temperature time series of SCOPE Climate, FYRE Climate, FYRE Daily and Safran over the case study cell for different periods and different time steps.





**Figure 10.** As for Fig. 9, but for precipitation.



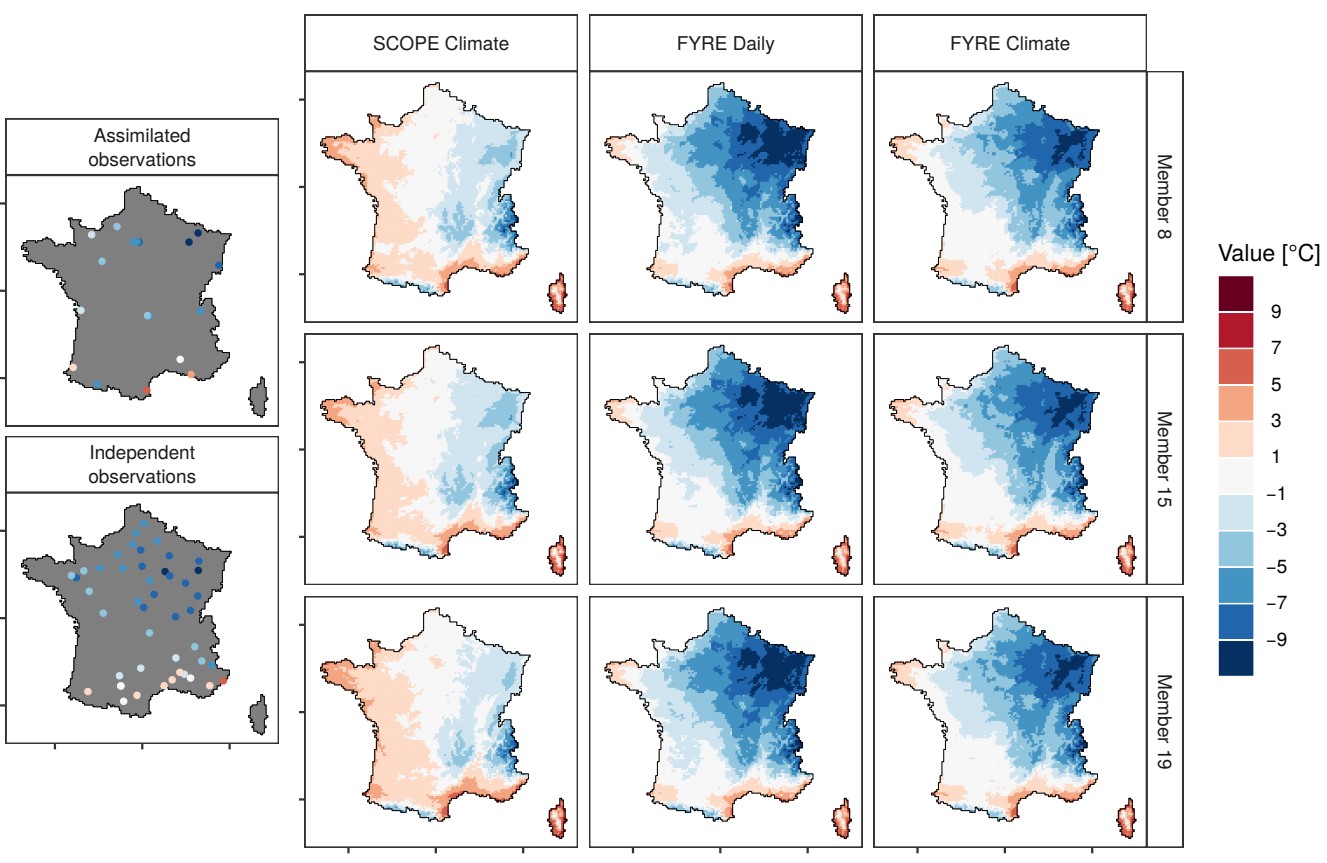

**Figure 11.** Average daily mean temperature in France during December 1879 from observations (left panels), and from three randomly selected members of SCOPE Climate/FYRE Daily/FYRE Climate (right panels).

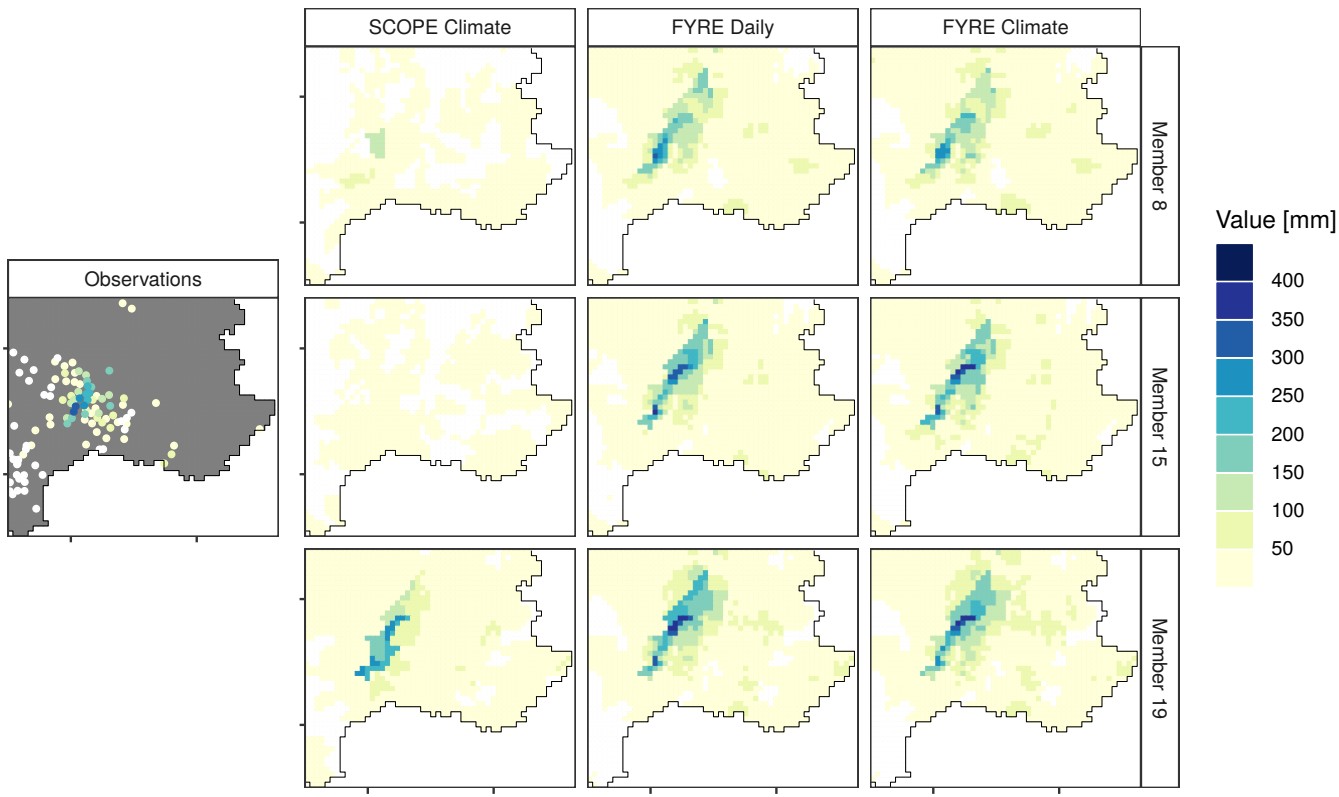

**Figure 12.** 21 September 1890 precipitation over South-East France from observations (left panel) and from three randomly selected members of SCOPE Climate/FYRE Daily/FYRE Climate (right panels).