# Peer review of "FYRE Climate: A high-resolution reanalysis of daily precipitation and temperature in France from 1871 to 2012"

_Climate of the Past, 2020_

## Referee Comment (RC1)

**Summary**

This is a review of "FYRE Climate: A high-resolution reanalysis of daily precipitation and temperature in France from 1871 to 2012" by Alexandre Devers, Jean-Philippe Vidal, Claire Lauvernet, and Olivier Vannier. The authors have generated a new high-resolution dataset for daily precipitation and temperature over France. The authors used a method from their other work—an offline data assimilation framework in which station observations are assimilated to the background given by the downscaling of lower-resolution reanalysis using the ensemble Kalman filter (EnKF). The authors comprehensively assessed the new high-resolution reanalysis and found that it largely improves over the lower-resolution reanalysis. I will first give some general comments that I would like to discuss with the authors and then give specific ones.

**General comments**

- In the offline data assimilation procedure, the background is given by the downscaling of a low-resolution reanalysis. Assimilating observations can reduce the error due to downscaling and thus, one can expect that FYRE climate improves over the SCOPE climate. An interesting result to me is that the FYRE climate is found to be better than the Safran reanalysis (Fig. 6 and 7), which has the same resolution to the FYRE climate and is also created using observations. What is the main reason?
- If Safran is used as background, should we expect a product that is better than FYRE?
- How is the performance of FYRE in comparison with other datasets with similar resolution, including gridded observational dataset such as E-OBS?

**Specific comments**

1) Line 57: Without checking the other paper, it is unclear to me what has been done in this work and what has been in that.
2) Eq. (1): the dimension of X prime is incorrect.
3) Eq. (2): the left-hand side should be $Y_i$
4) Line 173: dimension of $\epsilon_i$ incorrect, should be $\epsilon_i \in R^{m}$
5) Line 178: dimension of K is wrong and if H is written in this way, it should be a matrix and the dimension should be $m \times n$.
6) Line 180: This is misleading. R is given as you said in section 3.2. Because $\epsilon$ follows a given distribution, you won't need the expectation equation of Evensen (2003).
7) Eq. (7): $\rho$ is $m \times m$, the product of $PH^T$ is $n \times m$, we can only compute Schur product for two matrices with same dimension.
8) Line 255: what is the inverse error function?
9) Line 266: null precipitation? Do you mean zero total annual precipitation?
10) Eq. (12): Does $P\_daily[y,c]$ mean the sum of $P\_daily[d,c]$?
11) Line 320-322: So? You give a reason but have not well explained it.
12) Fig. 5: Are the results averaged over time? The bias has negative and positive values, does this have an influence on the averaged results?
13) Fig. 5: why use median rather than ensemble mean?
14) Line 341 and line 346-349: no plots for the correlation of FYRE daily in Fig. 6.
15) Fig. 5 and Fig. 6: Fig. 5 shows that FYRE climate is better than FYRE daily if Safran is used as reference, whilst Fig. 6 shows that FYRE daily is better than FYRE climate if SMR is used as reference. Fig. 6 also indicates that if SMR is used as reference, then FYRE daily and climate are better than Safran. Therefore, I am not convinced that FYRE climate is generally better than FYRE daily. It is reasonable that FYRE climate performs better than FYRE daily in terms of annual variation (because FYRE climate is constructed using FYRE yearly, which is created using annual observations). But for daily and monthly data FYRE daily can be better. Also, for extreme events.

16) Fig. 7 and line 364-366: It is interesting that the SCOPE climate also performs worse during this period.
17) Line 415: Does analysis always have a smaller ensemble spread than background?
18) Fig. 10: I don't understand why there is a large separation of FYRE daily precipitation ensembles (also Fig. 7). EnKF gives an analysis whose error is (approximately) Gaussian. Here, the separation of analysis ensemble indicates that analysis error follows a bimodal distribution.
19) Line 431: how large is the difference between ensemble members? Should say more on ensemble uncertainty.
20) Regarding extreme events: observations are vital for the representation of extreme events. Authors have shown that even the observational data from a small number of stations can make a large contribution. Additionally, a high resolution is essential for an accurate description of extreme events. The gridded data products give a value for a cell, which is averaged over a small domain. If the temporal scale of an extreme event is much smaller than the grid size, then the event will be naturally underestimated. In contrast, station observation is a point measure, which can accurately measure the weather condition at a single point. The authors may want to have a look at Hu and Franzke (2020; https://doi.org/10.1029/2020GL089624), which shows that gridded observational datasets perform better than reanalysis products in terms of extreme daily precipitation.
21) Line 506-509: I don't really understand this paragraph.

**Technical corrections**

- Line 222: than -> that

---

## Author Comment (AC1)

We would like to very much thank the anonymous referee #1 for reviewing our study and her/his constructive comments. Please find below the referee's comments in black font and the authors' response in blue font.

**Summary**

This is a review of "FYRE Climate: A high-resolution reanalysis of daily precipitation and temperature in France from 1871 to 2012" by Alexandre Devers, Jean-Philippe Vidal, Claire Lauvernet, and Olivier Vannier. The authors have generated a new high-resolution dataset for daily precipitation and temperature over France. The authors used a method from their other work—an offline data assimilation framework in which station observations are assimilated to the background given by the downscaling of lower-resolution reanalysis using the ensemble Kalman filter (EnKF). The authors comprehensively assessed the new high-resolution reanalysis and found that it largely improves over the lower-resolution reanalysis. I will first give some general comments that I would like to discuss with the authors and then give specific ones.

**General comments**

- In the offline data assimilation procedure, the background is given by the downscaling of a low-resolution reanalysis. Assimilating observations can reduce the error due to downscaling and thus, one can expect that FYRE climate improves over the SCOPE climate. An interesting result to me is that the FYRE climate is found to be better than the Safran reanalysis (Fig. 6 and 7), which has the same resolution to the FYRE climate and is also created using observations. What is the main reason?

Indeed, FYRE Climate seems to perform better than the Safran reanalysis which was built assimilating precipitation and temperature as well. However, there are many differences between the two reanalyses, among which the following two may explain these differences : (1) Safran is based on the strong hypothesis of climatically homogeneous zones (of 15 cells each on average, but with large variations across France with up to 50 cells for one zone, see Vidal et al., 2010), where values only depend on altitude, and not on the specific 8-km cell, and (2) Safran uses as a background vertical profiles from the ERA-40 global reanalysis and operational Météo-France analyses after 2002 (for temperature) and from climatological values (for precipitation) as mentioned in Section 2.4.1, so with a larger spatial information content compared to SCOPE Climate used by FYRE Climate as a background. FYRE Climate has therefore more assets to match the individual local stations composing SMR. Finally, as mentioned in the text (lines 335-337), the low performance of Safran with respect to temperature may also come from differences in the computation of the daily mean temperature (average of Tx and Tn for SMR, and average of hourly values for Safran).

- If Safran is used as background, should we expect a product that is better than FYRE?

It would not be possible to use the Safran reanalysis as a background in the proposed assimilation set-up. Indeed, the Safran reanalysis is deterministic and thus does not provide the ensemble needed for applying an ensemble Kalman filter approach. Furthermore, and surely most importantly, most of the observations assimilated in FYRE Climate have already been used to build the Safran reanalysis, which prevents designing a proper assimilation framework using Safran as a background.

- How is the performance of FYRE in comparison with other datasets with similar resolution, including gridded observational dataset such as E-OBS?

A comparison of FYRE Climate with E-OBS would be interesting but has not been conducted yet. This is particularly true as a new ensemble version of E-OBS has been recently created (Cornes et al., 2018). However, the comparison would be limited to the 1950-2010 period while the goal of FYRE Climate was to span the entire twentieth century. We can also expect FYRE Climate to perform better than E-OBS on the 1950-2010 period since many more observations were used to build FYRE Climate, as we can see when comparing Figure 1 of the manuscript and Figure 1 in Cornes et al. (2018). FYRE Climate uses as much as 4000 (resp. 2500) precipitation (resp. temperature) stations for

France while E-OBS 16.0 uses only 9600 (resp. 4080) precipitation (resp. temperature) stations for the whole of Europe. Figure 1 in Cornes et al., 2018 moreover shows that the number of observations considered by E-OBS in France is quite low as compared to other countries such as Germany or Sweden.

**Specific comments**

1) Line 57: Without checking the other paper, it is unclear to me what has been done in this work and what has been in that.

The sentence will be changed for: "The work of Devers et al. (2020a) developed and tested the DA scheme over a short period of time (2009-2012) with assimilated observation density reproducing the historical density at a few carefully selected points in time between 1871 and 2012, representative of the evolution of the observation network (1871, 1900, 1930 and 1950). This study applies here over the 1871-2012 period the scheme they developed in order to produce the full FYRE Daily reanalysis, composed of 25 members of daily precipitation and temperature at a 8 km resolution over France."

2) Eq. (1): the dimension of X prime is incorrect.

Indeed, we will replace it with R^{n \times N}.

3) Eq. (2): the left-hand side should be Y_i

As \epsilon is the entire matrix containing the *n* vector \epsilon_i with i= 1 to N, we believe that the proper notation is Y.

4) Line 173: dimension of \epsilon_i incorrect, should be \epsilon_i in R^{m}

Indeed, we will replace it with \epsilon_i in R^{m}.

5) Line 178: dimension of K is wrong and if H is written in this way, it should be a matrix and the dimension should be m \times n.

The observation operator H is indeed a matrix m \times n and its dimension will be added. The dimension of K will be changed to n \times m.

6) Line 180: This is misleading. R is given as you said in section 3.2. Because \epsilon follows a given distribution, you won't need the expectation equation of Evensen (2003).

This is right. This sentence will be removed.

7) Eq. (7): \rho is m \times m, the product of PH^T is n \times m, we can only compute Schur product for two matrices with same dimension.

The dimension of \rho is indeed the same as PH^T (n \times m), this will be changed.

8) Line 255: what is the inverse error function?

The inverse error function is defined as:
$$\mathrm{erf}^{-1}(z) = \sum_{k=0}^{\infty} \frac{c_k}{2k+1} \left( \frac{\sqrt{\pi}}{2} z \right)^{2k+1}$$
This will be added as a footnote in the paper.

9) Line 266: null precipitation? Do you mean zero total annual precipitation?

Yes, even if this case is likely to never happen (in France), we prefer to define how to handle it.

10) Eq. (12): Does P_daily[y,c] mean the sum of P_daily[d,c]?

Yes, it is correct.

11) Line 320-322: So? You give a reason but have not well explained it.

The following sentence will be added (in Section 4.2.1 when the issue is first mentionned): "Indeed, this difference in the computation lead to a difference in the estimation of the mean daily temperature when the diurnal cycle is not perfectly symmetric".

12) Fig. 5: Are the results averaged over time? The bias has negative and positive values, does this have an influence on the averaged results?

Yes, the scores are averaged over the 1960-2000 period as indicated in the caption. The average bias could indeed be zero over the entire period but potentially with years with strong positive and negative values. That is why we propose to evaluate the datasets with a temporal point of view in Fig. 6 and Fig. 7.

13) Fig. 5: why use median rather than ensemble mean?

We prefered to use the ensemble median as it is a more robust estimate of the central tendency of an ensemble.

14) Line 341 and line 346-349: no plots for the correlation of FYRE daily in Fig. 6.

The correlation of FYRE Daily are in fact hidden by the correlation of FYRE Climate, as the correlation at daily time step are almost the same in the two reanalyses. This will be added in Fig. 6 caption.

15) Fig. 5 and Fig. 6: Fig. 5 shows that FYRE climate is better than FYRE daily if Safran is used as reference, whilst Fig. 6 shows that FYRE daily is better than FYRE climate if SMR is used as reference. Fig. 6 also indicates that if SMR is used as reference, then FYRE daily and climate are better than Safran. Therefore, I am not convinced that FYRE climate is generally better than FYRE daily. It is reasonable that FYRE climate performs better than FYRE daily in terms of annual variation (because FYRE climate is constructed using FYRE yearly, which is created using annual observations). But for daily and monthly data FYRE daily can be better. Also, for extreme events.

The conclusions from Fig. 5 and Fig. 6 are correct. The difficulty here is to find an appropriate reference, which is always the main issue when building reanalyses that assimilate all observations. Homogenized series in SMR are at a monthly time step, and comparisons between FYRE Climate and FYRE daily can therefore only be done at monthly or annual time scales. And indeed Fig. 6 shows that FYRE daily is a bit closer to SMR than FYRE Climate. However, following figures show two major issues in FYRE Daily : Fig. 7 (and also Fig. 10) shows much more pronounced bimodal ensembles for FYRE Daily (at all time scales), and more importantly Fig. 8 shows inconsistent multidecadal variations of both temperature and precipitation in FYRE Daily with respect to SMR and EPC.

16) Fig. 7 and line 364-366: It is interesting that the SCOPE climate also performs worse during this period.

Indeed. The fact that SCOPE Climate also performs worse in this period could possibly be linked to the lack of surface pressure observations assimilated across Western Europe in 20CR as a result of WWII, as shown by Fig. 2 of Cram et al. (2015) in their description of ISPDv2.

17) Line 415: Does analysis always have a smaller ensemble spread than background?

Yes, most of the time. However, Fig. 10 shows that assimilating contradictory observations leads (quite rarely in our case) to a bimodal ensemble and thus a potentially higher spread in the analysis than the background.

18) Fig. 10: I don't understand why there is a large separation of FYRE daily precipitation ensembles (also Fig. 7). EnKF gives an analysis whose error is (approximately) Gaussian. Here, the separation of analysis ensemble indicates that analysis error follows a bimodal distribution.

The large separation is due to the assimilation of two stations with contradictory values, possibly due to measurement errors. It is right that in a ideal world where variables are Gaussian and observations are consistent, the analysis would lead to a Gaussian distribution. However, we deal here with daily precipitation whose distribution is (1) positive, (2) skewed, and (3) with a spike in zero. Note that we put an emphasis on this issue by applying a Gaussian anamorphosis prior to the assimilation, but this does not eliminate completely this issue. Moreover, and perhaps more importantly, measurement errors (coming e.g. from exposure like proximity to walls or trees) may easily lead to inconsistent values within a given grid cell, and consequently to a multimodal analysis.

19) Line 431: how large is the difference between ensemble members? Should say more on ensemble uncertainty.

The following sentence will be added: "The three randomly selected members are used to give an idea of the ensemble dispersion over the events". As we clearly see in Fig. 11 and Fig. 12, the dispersion in the reanalysis is lower than in the background, especially for precipitation.

20) Regarding extreme events: observations are vital for the representation of extreme events. Authors have shown that even the observational data from a small number of stations can make a large contribution. Additionally, a high resolution is essential for an accurate description of extreme events. The gridded data products give a value for a cell, which is averaged over a small domain. If the temporal scale of an extreme event is much smaller than the grid size, then the event will be naturally underestimated. In contrast, station observation is a point measure, which can accurately measure the weather condition at a single point. The authors may want to have a look at Hu and Franzke (2020, https://doi.org/10.1029/2020GL089624), which shows that gridded observational datasets perform better than reanalysis products in terms of extreme daily precipitation.

Thank you for the reference. However, the two reanalysis products considered in Hu and Franzke (2020) – ERA5 (Hersbach et al., 2020) and COSMO-REA6 (Bollmeyer et al., 2015) – do not assimilate precipitation observations (at least over the spatial domain studied by the authors). Hence their results, and all the more so for extreme values. On the contrary, FYRE Climate directly assimilate precipitation observations, so we can expect to reach values as high (or higher) as those observed at stations. And this is exactly what is shown in Fig. 12 (see also lines 461-463).

21) Line 506-509: I don't really understand this paragraph.

As it is strongly advised not to assimilate twice the same observations in a DA scheme (not to overemphasize observations with respect to the background), this paragraph discusses the fact that we combined FYRE Daily (assimilating daily observations) and FYRE Yearly (assimilating the same observations but at a yearly time step) into FYRE Climate. We then make the link with paleoclimate studies that applied similar set-ups..

**Technical corrections**

- Line 222: than -> that

We will replace it.

References

Bollmeyer, C., Keller, J.D., Ohlwein, C., Wahl, S., Crewell, S., Friederichs, P., Hense, A., Keune, J., Kneifel, S., Pscheidt, I., Redl, S. and Steinke, S. (2015), Towards a high-resolution regional reanalysis for the European CORDEX domain. Quarterly Journal of the Royal Meteorological Society, 141: 1-15, https://doi.org/10.1002/qj.2486

Cornes, R. C., van der Schrier, G., van den Besselaar, E. J. M. & Jones, P. D. (2018) An ensemble version of the E-OBS temperature and precipitation data sets, Journal of Geophysical Research: Atmospheres, 123, 9391-9409, https://doi.org/10.1029/2017JD028200

Cram, T. A., Compo, G. P., Yin, X., Allan, R. J., McColl, C., Vose, R. S., Whitaker, J. S., Matsui, N., Ashcroft, L., Auchmann, R., Bessemoulin, P., Brandsma, T., Brohan, P., Brunet, M., Comeaux, J., Crouthamel, R., Gleason Jr, B. E., Groisman, P. Y., Hersbach, H., Jones, P. D., Jónsson, T., Jourdain, S., Kelly, G., Knapp, K. R., Kruger, A., Kubota, H., Lentini, G., Lorrey, A., Lott, N., Lubker, S. J., Luterbacher, J., Marshall, G. J., Maugeri, M., Mock, C. J., Mok, H. Y., Nordli, Ø., Rodwell, M. J., Ross, T. F., Schuster, D., Srnec, L., Valente, M. A., Vizi, Z., Wang, X. L., Westcott, N., Woollen, J. S. & Worley, S. J. (2015) The International Surface Pressure Databank version 2. Geoscience Data Journal, 2, 31-46, https://doi.org/10.1002/gdj3.25

Hersbach, H., Bell, B., Berrisford, P., Hirahara, S., Horányi, A., Muñoz-Sabater, J., Nicolas, J., Peubey, C., Radu, R., Schepers, D., Simmons, A., Soci, C., Abdalla, S., Abellan, X., Balsamo, G., Bechtold, P., Biavati, G., Bidlot, J., Bonavita, M., De Chiara, G., Dahlgren, P., Dee, D., Diamantakis, M., Dragani, R., Flemming, J., Forbes, R., Fuentes, M., Geer, A., Haimberger, L., Healy, S., Hogan, R. J., Hólm, E., Janisková, M., Keeley, S., Laloyaux, P., Lopez, P., Lupu, C., Radnoti, G., de Rosnay, P., Rozum, I., Vamborg, F., Villaume, S. & Thépaut, J.-N. (2020) The ERA5 global reanalysis. Quarterly Journal of the Royal Meteorological Society, 146, 1999-2049, https://doi.org/10.1002/qj.3803

Hu, G., Franzke, C. L. E. (2020) Evaluation of daily precipitation extremes in reanalysis and gridded observation-based data sets over Germany. Geophysical Research Letters, 47, e2020GL089624. https://doi.org/10.1029/2020GL089624

Vidal, J.-P., Martin, E., Franchistéguy, L., Baillon, M. & Soubeyroux, J.-M. (2010) A 50-year high-resolution atmospheric reanalysis over France with the Safran system. International Journal of Climatology, 30, 1627-1644, https://doi.org/10.1002/joc.2003

---

## Author Comment (AC2)

We would like to thank the anonymous referee #2 for reviewing our study and her/his constructive comments. Please find below the referee's comments in black font and the authors' response in blue font.

General comments:

This paper describes a new reanalysis product over France for temperature and precipitation on an 8km grid during 1871--2012. A significant, and interesting, aspect of this work is the hybridization of results from ensemble data assimilation schemes for daily and annual timescales. This work builds upon a statistically downscaled 25-member ensemble from the 20th Century Reanalysis (SCOPE Climate). SCOPE provides the prior ensemble for both the daily and yearly reanalyses. Given such a small ensemble, covariance localization is critical, and seems to be well thought out in this study. There are many details to this work, most of which are well described. I have only one significant specific comment, and depending on how the editor chooses to guide the authors in revision, this could involve either involve minor or major revisions.

Specific comments:

Validation is an essential aspect of reanalysis studies like this one, and the authors have done a good job comparing against a high-resolution reference dataset (SMR). The problem as I see it is that the same observations have been used repeatedly for several aspects of this study, so that there is not real independent validation. Given the likely strong influence of the terrain function (equation 6), showing the sensitivity of the results to randomly removing a significant percentage of the observations (e.g., one third) from the data assimilation would be a good way to address this issue. It would also allow for validating against the withheld observations, including an estimate of how well the ensemble is calibrated by comparing the error of the ensemble mean to the ensemble spread for these withheld observations.

The purpose of this paper was to describe the creation and the features of the full FYRE Climate reanalysis. Hence, the paper included multiple comparison to other datasets (Safran, SMR and EPC) which are products mainly based on observation. However, as we can see in the Fig. 1 the observations are rather scarce and sparse before 1930. Note that this is particularly the case in mountainous areas. Thus, it is not possible to remove a third of the observations over the entire period and at the same time to have a strong network in validation. For example, in 1900 this would lead to a set of validation of ~300 stations for precipitation and ~25 for temperature without any station in the north or in the Alps (the main mountainous area in France). Moreover, it would be impossible to withhold the same set of stations for the whole period given the large changes in the network.

For all these reasons, and as we shared with the referee the need for a proper validation with independent observations, we therefore dedicated a full article (Devers et al., 2020a) where the DA scheme (identical to the one used in this article) is tested on a short and recent period (2009-2012). The amount of observations during this period allowed to remove a large amount of stationd to simulate the observation network as seen specifically in 1871, 1900, 1930 and 1950, both in terms of number and spatial coverage. Furthermore, this method allowed validating the DA scheme on numerous independent stations (783 for precipitation and 1500 for temperature) with a good coverage of mountainous area.

The validation procedure of the DA scheme included the use of the CRPS (Continuous Rank Probability Score) decomposition into Reliability (representative of the ensemble calibration) and Potential CRPS (representative of the accuracy). The results showed that the ensemble produced is well calibrated with a low Reliability component (close to the background Reliability) as well as a low Potential CRPS. Of course, this study showed that the performance of the reanalysis (mainly through the potential CRPS) decreases when the density of assimilated observations decreases, i.e. when one go further back in time (cf. Fig. 7 and Fig. 10 in Devers et al., 2020a).

We will therefore emphasize the added value of this preliminary but nevertheless required validation experiments in the introduction (line 57). The reader will thus be referred to more information on this crucial and difficult validation point shared by all reanalysis studies that precisely aim at using all available data at hand.

Minor comments and Technical corrections:

line 21: what are "discharge observations?"

The discharge observations are the amount of water going through the river section in a certain amount of time. For the sake of clarity, we will replace it by "streamflow observations".

line 59: SCOPE has not yet been defined

The acronym of SCOPE (Spatially COherent Probabilistic Extension Method) will be added before (line 47) and the corresponding reference (Caillouet et al., 2019) will be recalled here (line 59).

lines 75-80: This is meaningless jargon to most readers of CP. A clearer explanation for the general audience is needed in a background section.

We will try and simplify this paragraph as follows: "The SCOPE (Spatially COherent Probabilistic Extension Method, Caillouet et al., 2016, 2017) climate downscaling method is based on the analogue approach, which assumes that similar large-scale patterns of atmospheric circulation lead to similar local meteorological conditions of e.g. temperature and precipitation (Lorenz, 1969). SCOPE uses an ensemble analogue approach to reconstruct high-resolution climate fields from large-scale information on atmospheric circulation. SCOPE draws on several works on climate downscaling with analogues (Radanovics et al., 2013; Ben Daoud et al. 2016; Caillouet et al., 2016, 2017), and the reader is referred to these for more details. In short, based on information on large-scale atmospheric circulation from e.g. a global reanalysis, SCOPE generates an ensemble of high-resolution daily meteorological fields through a resampling of an archive of high-resolution meteorological fields. Note that the resulting fields from each ensemble member are coherent spatially as well as across variables, thanks to the use of the Schaake Shuffle (Clark et al., 2004)."

References

Caillouet, L., Vidal, J.-P., Sauquet, E., Graff, B. & Soubeyroux, J.-M. (2019) SCOPE Climate: a 142-year daily high-resolution ensemble meteorological reconstruction dataset over France. Earth System Science Data, 11, 241-260, https://doi.org/10.5194/essd-11-241-2019

Devers, A., Vidal, J.-P., Lauvernet, C., Graff, B. & Vannier, O. (2020a) A framework for high-resolution meteorological surface reanalysis through offline data assimilation in an ensemble of downscaled reconstructions. Quarterly Journal of the Royal Meteorological Society, 146, 153-173, https://doi.org/10.1002/qj.3663